# 7T MP2RAGE for cortical myelin segmentation: Impact of aging

**Susanne G. Mueller** *

Dept. of Radiology, University of California, San Francisco, San Francisco, CA, United States of America

* susanne.mueller@ucsf.edu

## Abstract

### Background

Myelin and iron are major contributors to the cortical MR signal. The aim of this study was to investigate 1. Can MP2RAGE-derived contrasts at 7T in combination with k-means clustering be used to distinguish between heavily and sparsely myelinated layers in cortical gray matter (GM)? 2. Does this approach provide meaningful biological information?

### Methods

The following contrasts were generated from the 7T MP2RAGE images from 45 healthy controls (age: 19–75, f/m = 23/22) from the ATAG data repository: 1. T1 weighted image (UNI). 2. T1 relaxation image (T1map). 3. INVC/T1map ratio (RATIO). K-means clustering identified 6 clusters/tissue maps (csf, csf/gm-transition, wm, wm/gm transition, heavily myelinated cortical GM (dGM), sparsely myelinated cortical GM (sGM)). These tissue maps were then processed with SPM/DARTEL (volume-based analyses) and Freesurfer (surface-based analyses) and dGM and sGM volume/thickness of young adults (n = 27, 19–27 years) compared to those of older adults (n = 18, 42–75 years) at p<0.001 uncorrected.

### Results

The resulting maps showed good agreement with histological maps in the literature. Volume- and surface analyses found age-related dGM loss/thinning in the mid-posterior cingulate and parahippocampal/entorhinal gyrus and age-related sGM losses in lateral, mesial and orbitofrontal frontal, insular cortex and superior temporal gyrus.

### Conclusion

The MP2RAGE derived UNI, T1map and RATIO contrasts can be used to identify dGM and sGM. Considering the close relationship between cortical myelo- and cytoarchitecture, the findings reported here indicate that this new technique might provide new insights into the nature of cortical GM loss in physiological and pathological conditions.

**Data Availability Statement:** The data used for this project can be found at https://www.nitrc.org/projects/atag_mri_scans/. The Matlab scripts used to generate the cortical segmentations have been made available on Mendeley Data

("cortical_myelin_segmentation", Mendeley Data, V1, doi: 10.17632/nrdmwdh633.1).

**Funding:** The author received no specific funding for this work.

**Competing interests:** The author has declared that no competing interests exist.

## Introduction

The cortical mantle is not homogeneous but can be subdivided into up to six distinctive layers or laminae based on cytoarchitectural features such as type, distribution, density, size, etc. of its neurons. This laminar organization shows regional variations [1] that can be used to subdivide the cortex into distinct cytoarchitectural and functional areas [2–4]. The layered structure also determines how these cortical areas are interconnected with each other and with subcortical structures. The most basic connectivity and receptor expression principles generalize across the cortical mantle [2–4]. Layer 2 and upper layer 3 for example contain cortico-amygdala/entorhinal-hippocampal projecting neurons and cortico-cortical feedback, i.e., higher-order to lower-order cortex projecting neurons, and lower layer 3 and 4 contain cortico-cortical feedforward or lower-order to higher-order cortex projecting and callosally projecting neurons. Layers 2–3 are also usually rich in GABA A and B, glutamatergic AMPA, NMDA, dopamine 1 and serotoninergic HT 5a receptors [4]. With exception of M1 and S1, glutamatergic kainate receptors are predominantly expressed in layers 5 and 6 [4]. Layer 5 contains cortico-brainstem, cortico-striatal, cortico-thalamic and cortico-cortical feedback neurons and layer 6 contains cortico-thalamic, cortico-claustral and cortico-cortical feedback neurons. In association cortices layers 5 and 6 contain cortico-cortical feedforward as well as feedback neurons that project to different areas though [2, 3]. This layer-specific functional organization is also the reason why the pathological hallmarks of some neurodegenerative diseases preferentially affect and spread via certain layers but spare others [5–7], e.g., there is evidence that tau pathology in Alzheimer's disease preferentially affects layers 5 and 6 and spreads via the feedback neurons in these layers [8].

Unfortunately, neither in vivo nor ex vivo magnetic resonance (MR) imaging has currently the necessary resolution to capture the cytoarchitectural features needed to distinguish between different cytoarchitectural layers. However, the research on cortical myelination patterns published by the Vogts and their coworkers shows considerable concordance between cyto- and myeloarchitecture [4, 9, 10]. Cortical myelin and iron bound to cortical myelin represent the main contributors to the cortical MR signal [11–13]. MR imaging of cortical myelin/iron might therefore provide valuable insights into the myeloarchitecture and by extension cytoarchitecture and functional cortical organization. These observations motivated several groups to explore the use of myelin sensitive and myelin/iron sensitive MR contrasts such as T1 weighted, T1 relaxation, R1, R2*, magnetization transfer imaging, and their combinations, for example T1 weighted/T2 weighted or T1 weighted/FLAIR imaging, for that purpose [e.g., 11, 14–21]. The study presented here investigated the usefulness of a combination of contrasts derived from a 7T MP2RAGE [22] for cortical myelin imaging. The MP2RAGE acquires two gradient echo images with different inversion times (INV1 and INV2] that are typically used 1. To obtain a T1-weighted image (UNI) free of proton density and T2* contrast and greatly reduced reception bias field and transmit field inhomogeneity and 2. To calculate a high resolution, whole brain T1 relaxation map (T1map) that has already been successfully used for cortical myelin mapping [13, 23]. Three additional contrasts were created by calculating ratio images from these outputs, an UNI/T1map ratio image (sRATIO), an INV1/INV2 ratio image (INVC) and an INVC/T1 map ratio image (RATIO). The INVC and RATIO images were of particular interest for this project because their cortical gray matter rim is divided into an inner low and an outer intermediate intensity zone whose thicknesses show regional variations consistent with heavily and sparsely myelinated cortical areas (cf. Fig 1). The first aim of this project was therefore to investigate if it is possible to use these two contrasts either alone or in combination with other MP2RAGE contrasts to segment these two cortical intensity zones. The second aim was to compare the resulting segmentations with histological myelin maps

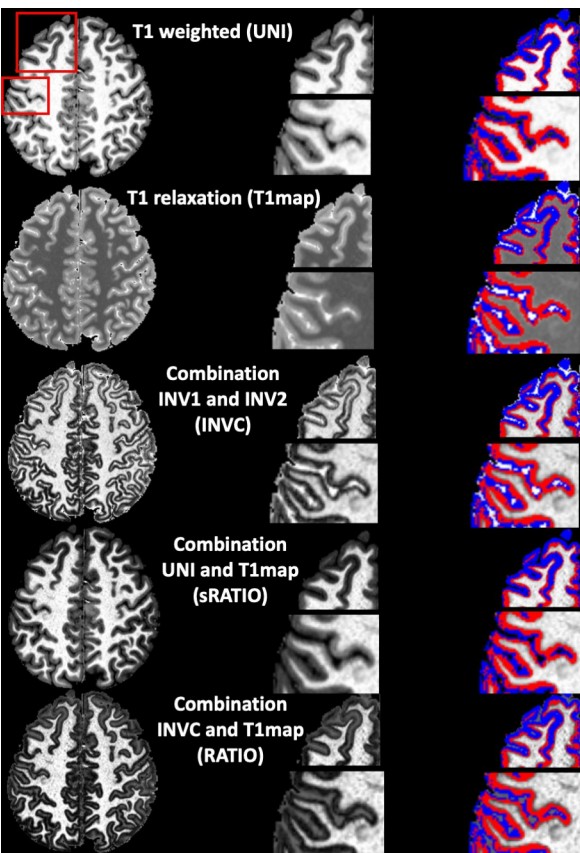

**Fig 1. Contrasts derived from the MP2RAGE of a 25 year old female subject.** INVC and RATIO show an inner low (red) and outer intermediate (blue) intensity cortical zone. First column shows an axial slice, middle column enlarged sections and the third column, the enlarged sections with corresponding segmentations.

from the literature [4] to better understand how the two zones relate to heavily and sparsely myelinated cortical layers. The third and last aim was to demonstrate that these segmentations capture biological meaningful information, i.e., to investigate if age related gray matter loss affects heavily and sparsely myelinated cortical layers differently.

## Methods

**Study population.** 45 data sets from the publicly available "Atlasing of The Basal Ganglia (ATAG) data repository (please see [24] for more detail). The completely anonymized data in the ATAG data repository was collected with the approval of their local ethics committees by researchers from the University of Amsterdam and Max Planck Institute for Human Cognitive and Brain Sciences [24] and is distributed under the CC01.0 Universal Public Domain Dedication license which allows free use and modification of the data set. Studies using anonymized data where it is not possible to ascertain the subject's identity are considered non-human subject studies and exempt from UCSF IRB review. The data sets used in this project were selected based on their imaging data quality (signal drop out/noise in inferior temporal lobe region, (S1 Table)). These 45 healthy subjects were divided into a "young adult" group (age range: 19–28 years, mean age 24.04 (2.488) years, f/m = 14/13) and an "older adult" group (age range: 42–75 years, mean age: 60.5 (9.8) years, f/m = 9/9).

## Imaging

All participants had undergone a structural scan on a 7T Siemens Magnetom MR scanner with a 24 channel Nova head coil. The whole brain MP2RAGE [22] with a TR = 5000 ms, echo time = 2.45 ms, inversion times TI1/TI2 = 900/2750 ms, flip angle 5 and 3 degrees, bandwidth = 250 Hz/Px, voxel size 0.7 mm isotropic and 240 sagittal slices and an acquisition time of 10.57 min was used for this project. The image processing steps are summarized in Fig 2.

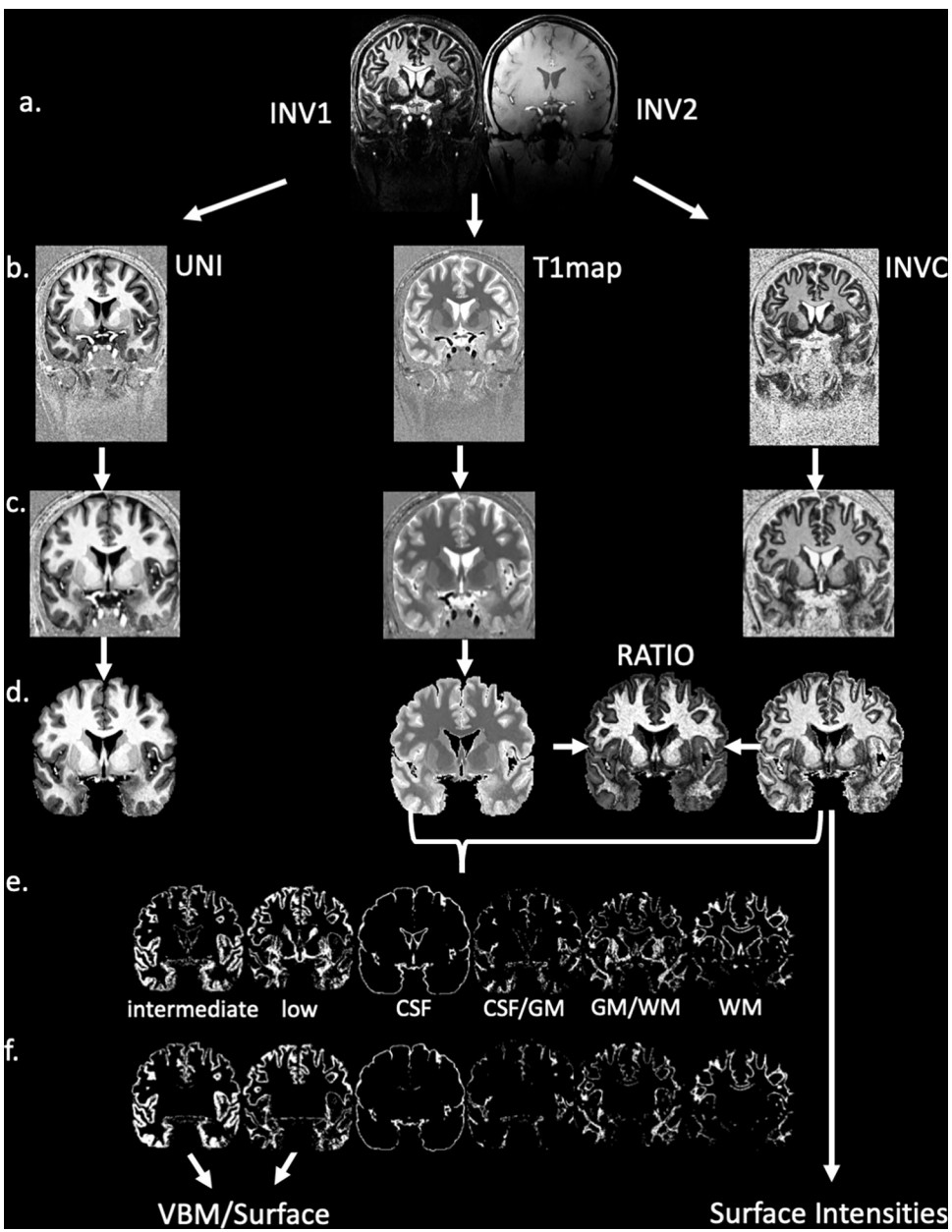

**Fig 2. Processing flow chart.** a. original INV1 and INV2 affected by the typical 7T field inhomogeneities. b. UNI, T1map and INVC map derived from different combinations of the INV1 and INV2. Bias fields etc. are greatly reduced compared to the two INV images. c. Spatial normalization into MNI space and correction of residual bias field with SPM "segment". d. Skull-striped UNI, T1 map and INVC. The two latter images are used to calculate the RATIO image (see text). e. Raw segmentations produced by k-means clustering. f. final segmentations after clean-up.

The MP2RAGE sequence generates an INV1 or TI1 gradient echo image, and an INV2 or TI2 gradient echo image (Fig 2A). These two images can be used to calculate three additional images (Fig 2B): 1. A synthetic T1 weighted image (UNI) that is derived from the complex INV1 and INV2 data. 2. A T1 relaxation map (T1map) that can be calculated from the INV1 and INV2 images with a protocol specific look-up table [22]. 3. INV1/INV2 ratio image (INVC) that is generated by combining INV1 and INV2 using the following formula:

$$INVC = \frac{INV1*INV2}{(INV1*INV1) + (INV2*INV2)} *100$$

Because the INV1 and INV2 are acquired under the same conditions B1$^-$, M$_0$, T2$^*$ are affected in an identical matter, and therefore the three images derived from combining them are largely free of proton density contrast, T2 contrast, reception bias field and transit field inhomogeneities.

## Image processing

**Pre-processing.**   The bias correction algorithm implemented in the SPM 12 routine "Unified segmentation" was used to remove the residual bias field from UNI, T1map and INVC image. Gray, white and CSF maps were generated from UNI with SPM12's "segment" [25]. The gray matter map was spatially normalized into MNI space using SPM12's "normalize" function while maintaining the original resolution, and the forward and inverse transformations of this step calculated. The former was applied to all outputs (bias corrected UNI, T1map, INVC, tissue maps, and a binary brain tissue mask derived by combining the normalized gray and white matter tissue maps). The brain tissue mask was used for skull-stripping the bias corrected and spatially normalized UNI, T1map and INVC (Fig 2C). The last step was to calculate two additional contrasts 1. A simple RATIO image (sRATIO) by dividing the skull-stripped UNI by the T1map and 2. A double RATIO image (RATIO) by dividing the skull-stripped INVC ratio image by the T1map.

**Cortical segmentation: Raw maps.**   The combination of image contrasts allowing for the best segmentation of the low and intermediate cortical intensity zones was determined based on the results in 4 randomly selected subjects aged 26, 27, 49 and 62 years. A region of interest map was generated by smoothing the UNI derived gray matter map with a Gaussian smoothing kernel of 2 by 2 by 2 mm FWHM and then binarizing it by thresholding at 0.15. This threshold resulted in a mask that not only contained the gray matter rim but also sulcal/gyral csf and subcortical white matter as well as csf/gray, gray/white transition or partial volume regions. The reasons for choosing this larger mask instead of just a mask focusing on the cortical rim were 1. The cortical rim borders are determined by the same segmentation procedure that is used for the other tissue types and independent from the performance of the SPM segmentation algorithm. 2. Although not of interest for this project, the gray/white partial volume map may contain important information for diseases characterized by gray/white blurring, e.g., subtle cortical malformations. This mask was used to extract the tissue intensities from each of the subject's images, that were converted into image specific z-scores and then alone or combined with the other contrasts supplied to the k-means clustering algorithm implemented in MATLAB 9.4 (The Math Works, Natick, MA; number of clusters $n = 6$, squared Euclidian distance function, maximum number of iterations = 1,000, replicates = 100). Table 1 lists the evaluated contrast combinations. The INVC and RATIO contrasts were characterized by visually discernible inner and outer gray matter rim zones. This was not the case for sRATIO image, but its tissue contrast behavior corresponds in many ways to that of the T1/T2 ratio image that is most often used for myelin mapping [16]. The INVC, RATIO and sRATIO images were therefore the only images that were evaluated re their ability to segment the six

**Table 1. List of evaluated contrast combinations.**

| Combination | csf | csf-gm | sGM | dGM | gm-wm | wm | Comment |
|---|---|---|---|---|---|---|---|
| INVC | yes | no | no | no | no | no | segmentation failure |
| sRATIO | yes | no | no | yes | no | yes | segmentation failure |
| **sRATIO, INVC** | yes | yes | yes | yes | yes | yes | Pass |
| **sRATIO, T1map** | yes | yes | yes | yes | yes | yes | Pass |
| sRATIO, T1map, INVC | yes | yes | yes | yes | yes | yes | noisy sGM segmentation in old |
| T1map, INVC | yes | yes | yes | yes | yes | yes | noisy sGM segmentation in old |
| **UNI, INVC** | yes | yes | yes | yes | yes | yes | Pass |
| UNI, T1map, INVC | yes | yes | yes | yes | yes | yes | noisy sGM segmentation in old |
| UNI, T1 map, sRATIO | yes | yes | yes | yes | yes | yes | prominent outer image edges/dura residuals in dGM |
| RATIO | no | no | no | no | no | no | segmentation failure |
| RATIO, INVC | no | no | no | no | no | no | segmentation failure |
| RATIO, sRATIO | yes | no | no | no | no | yes | segmentation failure |
| **RATIO, T1map** | yes | yes | yes | yes | yes | yes | Pass |
| **RATIO, T1map, INVC** | yes | yes | yes | yes | yes | yes | Pass |
| **RATIO, T1map, sRATIO** | yes | yes | yes | yes | yes | yes | Pass |
| **RATIO, T1map, UNI** | yes | yes | yes | yes | yes | yes | Pass |
| RATIO, UNI | yes | no | no | yes | no | no | segmentation failure |
| RATIO, UNI, INVC | yes | yes | yes | yes | yes | yes | segmentation failure |
| RATIO, UNI, sRATIO | yes | no | no | yes | no | no | segmentation failure |

csf, csf segmentation, csf-gm, csf/gray matter partial volume, sGM, intermediate intensity gray matter rim zone in INVC

dGM, low indensity gray matter rim zone in INVC, gm-wm, gray/white partial volume zone, wm, white matter

**bold**, contrast combinations that were further evaluated, yes, tissue type can be identified, no, tissue type not identified

tissue types without additional contrasts. Contrast combinations in which the contrast behavior of all images was similar, e.g. the combination sRATIO and UNI, were not evaluated. The optimal number of clusters n = 6 was chosen based on the observation that the mask included 6 tissue types in the INVC (csf, white matter (wm), csf/gray matter partial volume (csf-gm), white/gray matter partial volume (gm-wm) and two gray matter contrasts corresponding to the low (dGM) and intermediate intensity region (sGM) in the INVC cortical rim). The correctness of 6 as the optimal cluster number was confirmed experimentally by exploring the range from 6 to 8 clusters in one young and one old subject. The cluster centroid information of each subject was matched to the centroid information of a randomly selected reference subject and the cluster numbering accordingly changed to ensure a consistent cluster numbering/centroid assignment across the four subjects for each of the contrast combinations. The visual assessment of the segmentation outputs was based on how well the contrast combination was able to identify the 6 tissue types in all four subjects.

Combinations passing the visual assessment where then further evaluated by calculating the mean distance between centroid and each voxel assigned to this cluster (within cluster distances) and the mean distance between each voxel and all other cluster centroids (between cluster distances). Using these two metrics, a contrast combination indicating a good separation between the voxel intensities is characterized by small within cluster distances and large between cluster distances.

**Cortical segmentation: Final maps.** The first pass segmentations generated with the optimal image combination underwent two clean-up steps to generate the final segmentations. 1. Due to the nature of the region of interest map, the first pass cluster images contained non cortical rim gray matter, e.g. cerebellum, thalamus, striatum, hippocampus, amygdala etc.. These regions were masked out by warping a binary cerebellum-subcortical map in MNI space onto

each subject's cluster maps. 2. Some of the intermediate intensity cortical rim voxels in regions that were affected by B1 inhomogeneities due to their closeness to the air-filled spaces, i.e., inferior temporal and orbito-frontal regions, were misclassified as partial volume csf/gray matter voxels and were reclassified as intermediate intensity cortical voxels if their probability of being gray was greater than 95% in the gray matter tissue map (Fig 2F).

## Application

Effects of aging on low and intermediate intensity cortical layers were investigated using t-tests (young adults > older adults) with surface- and volume-based analyses to allow for a comparison of the thinning/volume loss patterns and T peaks of the two approaches. Surface-based analyses have been shown to have a higher statistical sensitivity than volume-based analyses [26, 27]. This is explained by a higher between subject registration precision of surface-based approaches that match cortical folding patterns compared to volume-based approaches that match regional voxel-intensities. However, in contrast to the gray matter probability or gray-scale maps traditionally used for volume-based registration, the six final maps contain information about regionally specific intracortical features and account for partial volume effects. These features should improve registration precision and sensitivity of volume-based registration. Using a surface and a volume based approach allowed to investigate this issue. Finally, the bias corrected INVC image was used to investigate age-related myelin signal changes at depths of 0.25, 0.5 and 0.75 of the cortical thickness which is a commonly used analysis approach by studies investigating myelination differences with T1/T2 ratio images.

**Surface-based processing.** The bias-corrected UNI images were processed with Freesurfer 7.3.2 recon-all with the high resolution flag to maintain the submillimeter resolution and the no-ants-n4 flag to suppress the redundant n4 bias correction. Freesurfer's bbregister with nearest neighbor interpolation was used to co-register each subject's bias corrected UNI image to the brainmask image generated by Freesurfer. The resulting transformations were entered into the mris_preproc routine that was run with the projfrac flag to sample the image intensities of the INVC images at 0.25, 0.5 and 0.75 of the cortical rim and project them onto left and right hemispheric surfaces. Mri_vol2surf with the bb register transformations and projfrac-avg flag set to 0 1 0.1 followed by surf2surf and mri_concat with the final low and intermediate intensity cortical rim maps as inputs were used to calculate left and right low intensity and intermediate cortical rim thickness surfaces. After smoothing the outputs with a Gaussian smoothing kernel of 1 by 1 by 1 mm FWHM mri_glmfit and fspalm (5000 permutations) [28, 29] were used to compare INVC intensities and low intensity and intermediate cortical rim thickness of the "young adult group" with those of the "older adult group" at p = 0.001 uncorrected.

**Voxel-based morphometry.** The six cluster images from all 45 subjects were used as inputs for DARTEL's create template algorithm in SPM12 to generate a 6 cluster population template. The resulting transformations were applied to the cluster maps (no modulation) that were then smoothed with a Gaussian smoothing kernel of 1 by 1 by 1 mm FWHM. The smoothed cluster maps corresponding to the low intensity cortical rim and the intermediate intensity cortical rim of the "young adult" group were compared with those of the "older adult" group using SPM and the Threshold-Free Cluster Enhancement (TFCE) toolbox (http://dbm.neuro.uni-jena.de/tfce) (5000 permutations) at p = 0.001 uncorrected.

## Results and discussion

### Cortical segmentation: Development

**Selection of optimal image combination.** Table 1 shows the results of the visual assessment of the segmentation quality. Even though the two cortical zones were discernible on the

**Table 2. Within and between cluster distances.**

| Combination | distance type | csf | csf-gm | sGM | dGM | gm-wm | Wm | mean |
|---|---|---|---|---|---|---|---|---|
| **sRATIO, INVC** | within | 0.000 | 0.305 | 0.152 | 0.165 | 0.194 | 0.293 | 0.185 |
| | between | 5.723 | 5.192 | 2.354 | 2.626 | 3.012 | 6.766 | 4.279 |
| **sRATIO, T1map** | within | 0.000 | 0.159 | 0.058 | 0.062 | 0.087 | 0.174 | 0.090 |
| | between | 8.264 | 5.332 | 2.757 | 2.038 | 2.876 | 5.920 | 4.531 |
| **UNI, INVC** | within | 0.000 | 0.337 | 0.166 | 0.157 | 0.144 | 0.194 | 0.166 |
| | between | 7.057 | 5.333 | 2.235 | 2.691 | 3.008 | 5.368 | 4.282 |
| **RATIO, T1map** | within | 0.000 | 0.421 | 0.087 | 0.120 | 0.214 | 0.198 | 0.173 |
| | between | 8.330 | 7.069 | 2.505 | 2.429 | 2.602 | 4.495 | 4.572 |
| **RATIO, T1map, INVC** | within | 0.000 | 0.456 | 0.191 | 0.215 | 0.299 | 0.569 | 0.288 |
| | between | 11.490 | 7.857 | 3.612 | 3.950 | 3.643 | 9.323 | 6.646 |
| **RATIO, T1map, sRATIO** | within | 0.000 | 0.253 | 0.133 | 0.190 | 0.347 | 0.604 | 0.254 |
| | between | 10.894 | 5.990 | 3.666 | 3.507 | 4.644 | 11.731 | 6.739 |
| **RATIO, T1map, UNI** | within | 0.000 | 0.266 | 0.148 | 0.177 | 0.294 | 0.479 | **0.227** |
| | between | 12.225 | 6.275 | 3.586 | 3.596 | 4.568 | 10.161 | **6.735** |

INVC and RATIO image, the segmentation outputs generated from each of these images alone were not able to clearly segment the 6 tissue types in all four subjects. The same was true for five additional image combinations of which none included the T1map contrast. Combinations of sRATIO, T1map with INVC, and of T1map with INVC performed reasonably well with the images from the younger subjects but produced noisy gray matter rim segmentations in the oldest subject. Using a combination of UNI, T1 map and sRATIO as input image produced good segmentations of all 6 tissue categories. However the map corresponding to the low intensity region of the cortical rim also included voxels belonging to the residual dura at the image edge that were most prominent in the basal regions and could not reliably be eliminated with standard morphological image processing operations.

Table 2 lists the within and between mean cluster distances of the seven contrast combinations that passed the visual segmentation assessment. Using lowest within and largest between cluster distance of the mean (over all tissue categories) and of low (dGM) and intermediate (sGM) cortical rim intensities as a selection criterion, the combinations RATIO, T1map, sRATIO and RATIO, Tmap, UNI perform best with the latter having slightly better mean (over all tissue categories) values than the former. Based on this, the combination of RATIO, T1map and UNI was chosen to process all data sets.

**Characterization of cortical rim clusters.** The paper from Palomero-Gallagher and Zilles [4] provides detailed cyto- and myeloarchitecture information of 8 Brodmann Areas. Fig 3 shows a side-by-side comparison of the cyto- and myeloarchitecture of seven regions (precentral gyrus or M1, postcentral gyrus or S1, calcarine cortex or V1, extrastriate cortex, Brodmann area 44, 7, and p24) from Palomero-Gallagher and Zilles [4] with the corresponding MRI in which the low intensity (red) and intermediate intensity zones (blue) of the cortical rim have been segmented. The myelin staining clearly distinguishes between a bright or sparsely myelinated zone (layers 1–3) and a dark or heavily myelinated zone (layers 4–6) within the cortical rim. The low intensity/high intensity segmentations match the different thickness of these zones quite well suggesting that they might indeed represent sparsely and heavily myelinated layers.

## Cortical segmentation: Application

**Effects of aging on myeloarchitecture.** Fig 4. shows the findings of the surface and volume based analysis. Surface and volume losses affect the same regions although they tend to be

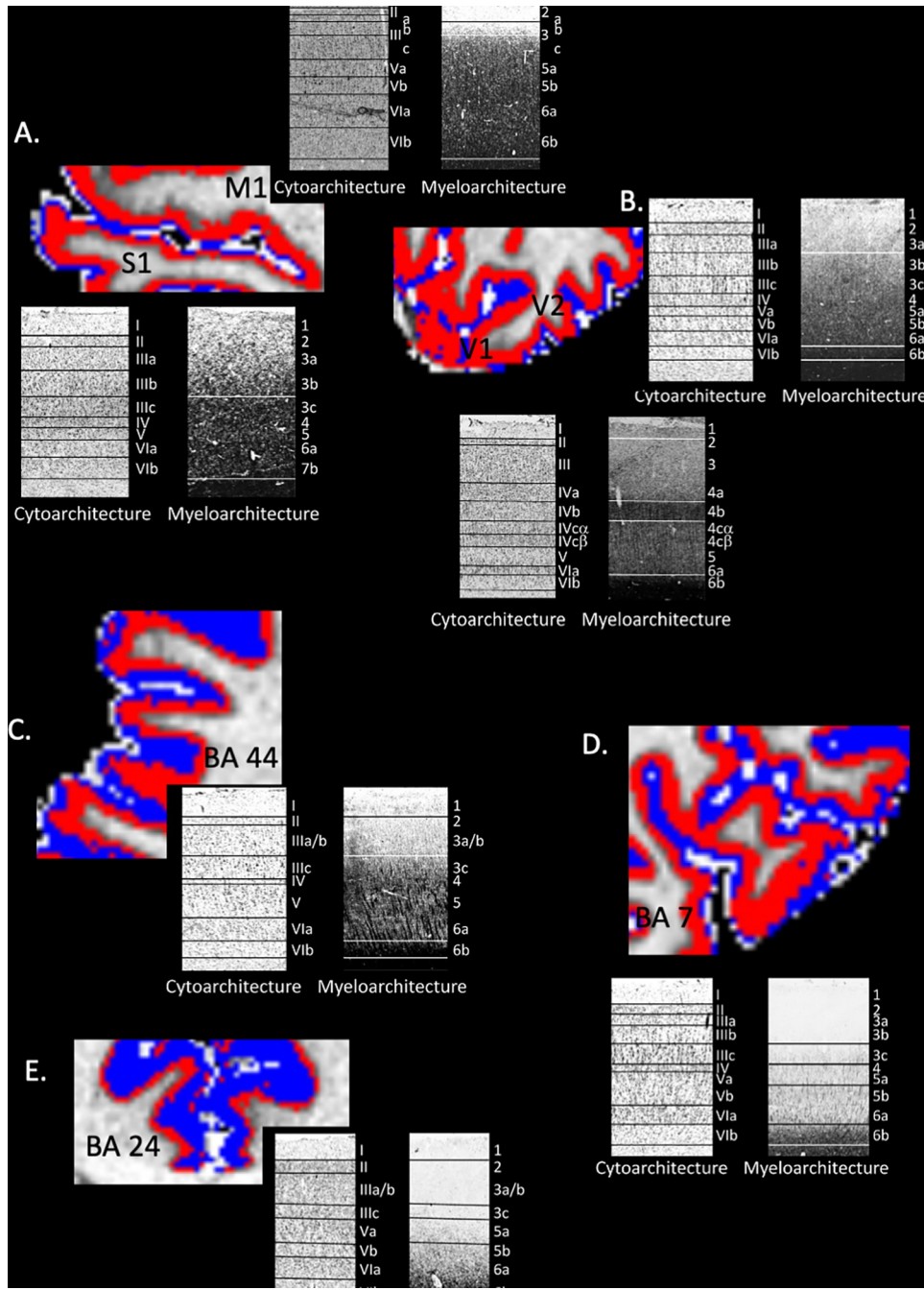

**Fig 3. Comparison between k-means segmentation of sparsely (blue) and heavily (red) myelinated cortical zones overlaid on the INVC image and histological preparations.** 2.a. shows the pre- (M1) and post- (S1) central gyrus. The former is about twice as thick as the latter which is consistent with the known morphology and is characterized by a prominent heavily myelinated zone with a thinner sparsely myelinated zone. The heavily myelinated zone of the S1 is thinner. 2.b. shows part of the V1 and V2. The Gennari stripe is not resolved but the heavily myelinated zone of the V1 appears to be thicker than of V2. 2. c. shows a section of Brodmann Area 44. 2.d. depicts a section of Brodmann Area 7. Its low intensity and intermediate zones correspond well to the myelination gradient seen in the histological preparation. 2. e. shows a section from Brodmann Area 24. Again, the outer intermediate and inner low intensity zones correspond well to the gradient seen in the histological preparation. The histological preparations used in this figure are taken from Zilles and Palomero-Gallagher [4] published under a CC BY 4.0 License.

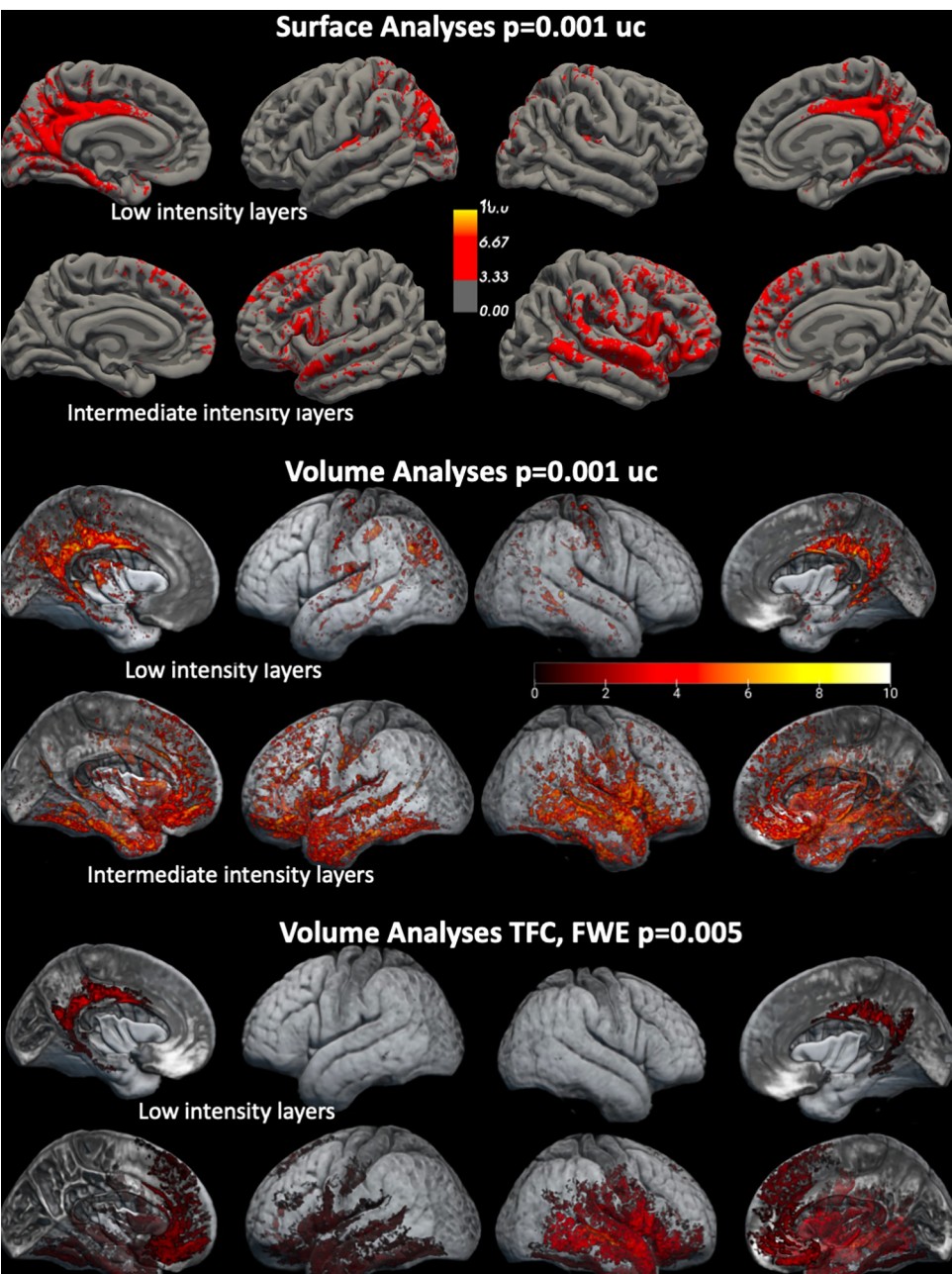

**Fig 4. Effects of aging on cortical layers.** Findings of surface-based analysis (upper third of panel) and volume-based analyses (middle third of panel). The intensity bars represent T scores. T maps were thresholded at -3, i.e., p = 0.001 and -10, i.e., p = $1.0e^{-10}$ The lower third of the panel shows the findings of the volume-based analyses after correction for multiple comparisons. Please see methods section for details regarding statistics.

somewhat more widespread in the volumetric analysis. Age related low intensity cortical layers thinning/volume losses are most prominent in the posterior cingulate/retrosplenial cortex, entorhinal/parahippocampal gyrus and posterior insular cortex bilaterally and left parietal inferior region.

Age related intermediate intensity cortical layers thinning/volume losses are most prominent in orbitofrontal, medial, inferior and lateral temporal, anterior insula, medial and inferior

lateral prefrontal regions. The peak T values of the volume-based analyses were higher than those for the surfaced-based analyses, particularly in the right middle/superior temporal gyrus and opercular region. Correcting for multiple comparisons (FWE p = 0.005, threshold-free cluster enhancement) in the volume-based analysis shows the same distribution of layer thinning/volume loss.

Fig 5. shows the age-related intensity changes in the INVC image. Displaying regions in which younger adults had significantly lower intensities than older adults at 0.5 and 0.75 highlighted the same regions as the comparison using low intensity cortical layer segmentation maps. Displaying regions in which younger adults had significantly higher intensities than

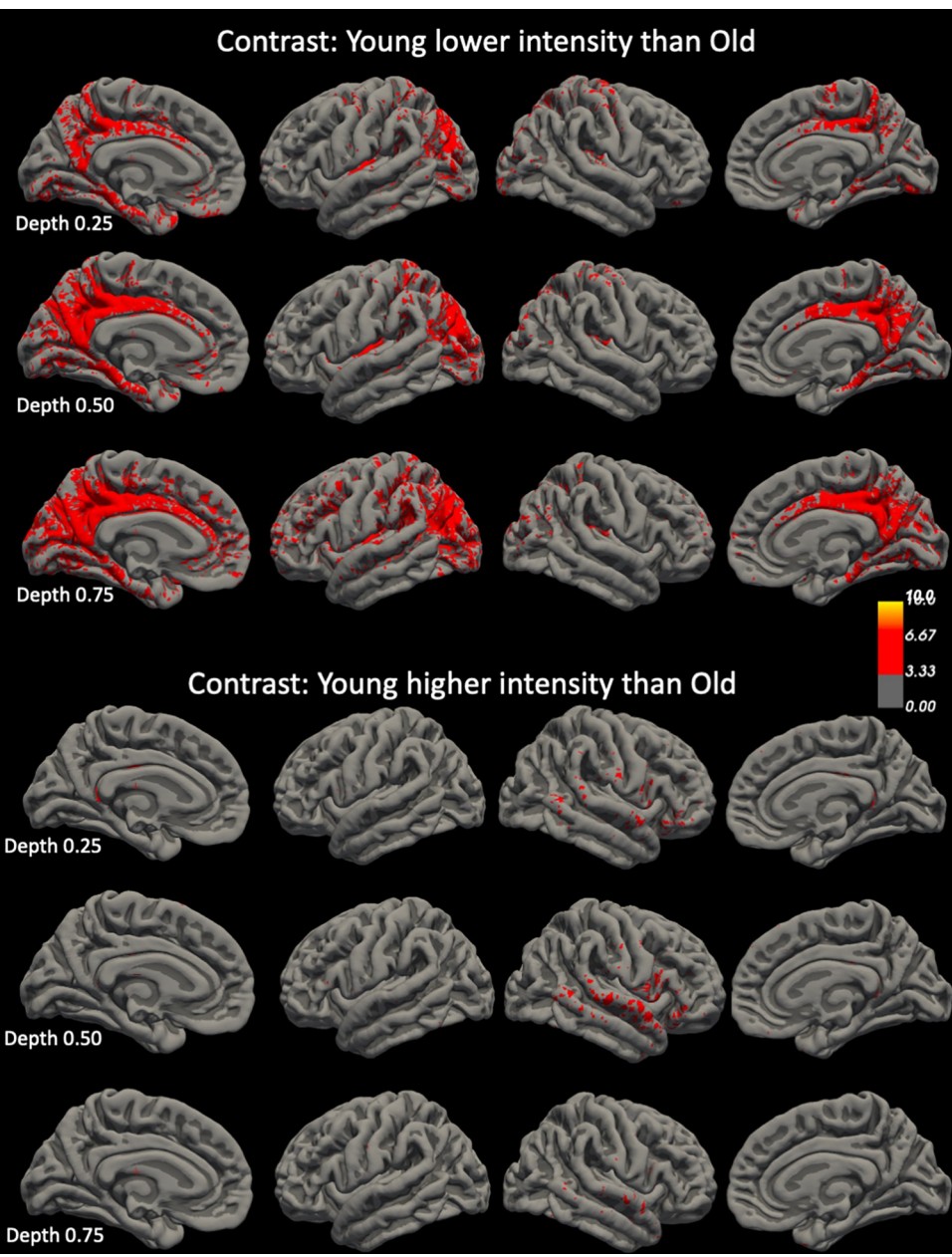

**Fig 5. Age-related differences of INVC intensities at different cortical depths.** The intensity bars represent T scores. T maps were thresholded at -3, i.e., p = 0.001 and -10, i.e., p = $1.0e^{-10}$.

older adults highlighted regions in the right middle and superior temporal gyrus at 0.5 and 0.25, i.e., differences indicating thinning of intermediate intensity INVC regions were only found in a fraction of the regions that were identified as having age-related thinning/volume loss in surface- and volume-based analyses with intermediate cortical layer segmentation maps as inputs.

## Conclusions

The study had two major findings: 1. The MP2RAGE sequence acquires two gradient echo images with different inversion times that can be used to generate five different contrasts: a T1 weighted image or UNI, a T1 relaxation map or T1map, a ratio image derived from the UNI and T1map image (sRATIO), a ratio image derived from the two gradient echo images (INVC) and a ratio image derived from the INVC and T1map (RATIO). The cortical gray matter rim of the latter two contrasts is divided into an inner low and an outer intermediate intensity zone whose thickness shows regional variations. K-means clustering with the UNI, T1map and RATIO images as input can segment these two cortical zones. The two zones correspond well to sparsely and heavily myelinated layers seen in histological preparations. Due to the close relationship between myelo- and cytoarchitecture this method has the potential to provide layer-specific cortical volumetric information and thus ultimately to provide insights into layer-specific physiological and pathological processes. 2. Voxel-based morphometry and surface-based thickness measurements both found the same pattern of layer-specific cortical gray matter volume loss/thinning in healthy aging. Healthy aging affects predominantly sparsely myelinated layers in frontal and temporal lobes and predominantly heavily myelinated cortical layers in posterior cingulate/retrosplenial cortex and parahippocampal/entorhinal regions.

Based on these findings, it is concluded that a combination of the contrasts derived from the two MP2RAGE inversion recovery images allows segmenting the cortical rim into maps representing sparsely and heavily myelinated cortical layers that convey biologically meaningful volumetric information that cannot be obtained with standard voxel-based morphometry or thickness measurements. The next paragraphs will discuss the proposed method and the findings in more detail.

As mentioned in the introduction, there is great interest in measuring intracortical myelin in vivo employing myelin sensitive MR contrasts such as T1, R1, R2*. Many of these studies are qualitative, i.e., focus on depicting a specific feature, e.g., the stripe of Gennari in the primary visual cortex (V1) or the transition between primary motor cortex (M1) to sensory motor cortex (S1) [11, 14, 20, 30, 31] while others attempt to quantify the myelin signal. Due to the laminar organization of the cortical rim, the latter is not trivial. A commonly chosen solution to this problem is to transform the cortical rim into surfaces and average the signal between the outer and inner surface (average approach) or to sample it at different depths of the cortical rim, e.g., 0.25, 0.5 and 0.75 of its thickness, and, if sampled often enough, to generate transcortical intensity profiles (profile approach) [e.g. 12, 16, 32–37]. Both approaches have limitations. The average approach only allows to distinguish between heavily and sparsely myelinated regions and provides no layer-specific information. The profile approach mimics the photometric curves that were introduced by Braitenberg [38] and Hopf [39] to describe the laminar myelination pattern in histological preparations. Its main limitation -at least with the common 0.25/0.5/0.75 approach- is that it might miss more subtle abnormalities in regions where either heavily or sparsely myelinated layers make up most of the cortical rim so that all intensity samples come from these layers. How this can influence the findings is demonstrated in this study where the 0.25/0.5/0.75 approach was used for the analysis of the INVC intensity images. Whereas this approach was able to detect the age-related thinning of the low-intensity

layers in the posterior cingulate/precuneus and parahippocampal regions where the thickness differences between heavily and sparsely myelinated layers are smaller, it missed the age-related changes in the prefrontal/left temporal regions where sparsely myelinated layers make up most of the cortical rim. Sampling the cortical rim intensities at more than 3 levels, e.g. 10 as was done by Sui et al [40], generates more detailed intensity profiles. However, it also necessitates the introduction of a measure to describe the profile which Sui et al. solved by calculating the root-mean-square deviation of the intensity profile from a linear regression or nonlinearity index (NLI). Selective thinning of one layer, e.g., the sparsely myelinated layer, will change the NLI. The approach used by Sui et al. resulted in a similar aging pattern as found in this study but the interpretation of a changing NLI is less intuitive than the thinning/volume loss of the layer maps used here.

Surface-based and volume-based analyses using cortical low- and intermediate intensity maps as inputs identified very similar patterns of age-related thinning/volume loss. The volume-based analyses had even higher T peaks than the surface-based analyses indicating at least a similar if not higher sensitivity of the volume-based analyses as surface-based analyses. These findings suggests that the additional biological features in the six cluster maps could indeed translate into an improved between subject-co-registration and thus higher sensitivity of volume-based analysis approaches.

The approach chosen in this study uses k-means clustering to divide the cortical rim into two sharply defined intensity zones that are assumed to correspond to the inner heavily and the outer sparsely myelinated cortical layers. The resulting maps can be used to investigate physiological and pathological changes of the cortical myelination pattern. However, dividing the cortical rim in just two zones is of course an oversimplification. While layers 5 and 6 are always heavily and layers 1 and 2 always sparsely myelinated across the cortical sheet, the same is not true for layers 3 and 4. The transition between layers 2 to 5 can either be relatively abrupt and layers 3 and 4 thus be either heavily, e.g. M1, S1, or sparsely myelinated, e.g., BA24, or more gradual with layers 3 and 4 showing an intermediate degree of myelination, e.g., V1 and BA 7 (cf Fig 3 and 5 [38]]. The different degrees of myelination and by extension the range of different intensities of layers 3 and 4 across the cortical sheet is too large and the image resolution of the ATAG data set too low as that the k-means clustering approach would be able to segment these two layers reliably from the other layers. Depending on their overall intensity range layers 3 and 4 are therefore either assigned to the sparsely myelinated zone, i.e., lumped together with layers 1 and 2, or to the heavily myelinated zone, i.e., lumped together with layers 5 and 6. This limitation has to be kept in mind when interpreting the findings obtained with this method.

The new layer specific approach's potential to provide biological meaningful information was assessed by mapping layer-specific gray matter thinning/volume loss in older adults compared to younger adults. Gray matter losses in sparsely myelinated layers affected temporo-lateral, temporo- inferior and medial, insular and orbitofrontal and dorsolateral prefrontal regions while gray matter losses in heavily myelinated layers were found in the posterior cingulate/retrosplenial cingulate and parahippocampal/entorhinal cortex. The prefrontal, insular and temporal cortices with the exception of the primary auditory cortex, inferior temporal and posterior insular cortices [9, 35] belong to the sparsely myelinated cortices indicating that the age-related losses likely affect layers 1–3 and eventually 4. These findings are consistent with the age-related prefrontal and temporal abnormalities described in primates and humans, i.e., loss of synapses in layers 2 and 3, loss of myelin in layer 4 and age-related loss of GABA-ergic and dopaminergic tonus in layers 2 and 3 that results in an abnormal feedforward inhibition [41–46]. The posterior cingulate/retrosplenial cortex and entorhinal/parahippocampal cortex belong to the more heavily myelinated regions indicating that age-related volume losses there

likely affect layers 5–6 and eventually 4 [38]. The entorhinal/perirhinal cortex is the first cortical region to accumulate neurofibrillary tangles in normal aging [47, 48] but also in Alzheimer's disease [8]. It is reciprocally connected to the parahippocampus and posterior cingulate/retrosplenial cortex which suggests an impaired feedback connection between these structures and the entorhinal/perirhinal cortex. The entorhinal/parahippocamal cortex and posterior cingulate/retrosplenial cortex project also reciprocally to the dorsolateral, orbitofrontal and medial frontal cortex, superior temporal gyrus and insula, i.e., the same regions that show age-related layer 2–4 volume losses [49–52]. This suggests that feedback as well as feedforward information exchange between limbic and prefrontal/temporal structures is altered in older adults. Taken together, the complex, regionally heterogeneous layer specific abnormalities in older adults correspond well to the known age related changes [41] in episodic memory (posterior cingulate, parahippocampus, entorhinal layer 5 and 6 atrophy) and working memory, attention, and inhibitory control (prefrontal layer 1–3 atrophy).

There are two previous studies that also attempted to distinguish between sparsely and heavily myelinated cortical layers using different approaches. Rowley et al. 2015 [21] acquired four T1 weighted 3T BRAVO sequences with 1 mm resolution (2 for each hemisphere that were co-registered and averaged to increase contrast/noise) and a proton-density weighted image with 1 mm resolution that was co-registered to the averaged T1 weighted image and then used to calculate a T1/PD ratio image with greatly reduced B1- and B1+ inhomogeneity. Fuzzy c-clustering with probabilistic segmentation with arbitrarily chosen thresholding at 0.5 for the gray matter regions and 0.1 for the white/gray boundary was used to identify 3 tissue clusters (white matter, myelinated GM and unmyelinated GM) in the T1/PD ratio image. Viviani et al. [15] acquired a multi-echo MPRAGE and FLAIR sequence with a 1 mm isotropic resolution. The images were co-registered and a FLAIR/MPRAGE ratio image calculated. These three images were used as input for the unified tissue segmentation algorithm implemented in SPM12 [25] whose Gaussian mixture modelling for gray matter was set to detect two intensities instead of the default one which split the gray matter rim into an inner and an outer rim. The two approaches have similarities with the one proposed here, e.g., use of multiple contrasts and a RATIO image, but also differences. The most notable besides the different segmentation approaches (fuzzy clustering, Gaussian mixture modelling vs. k-means clustering) is probably the need to co-register the different contrasts and the lower image resolution which will impact the detection of the transition between the two zones, particularly in regions with a thinner cortical rim (<4 mm). A meaningful comparison between the three approaches is difficult though because it would require data that was acquired in the same population with the same resolution and ideally be combined with ground truth information, i.e., histological data.

The study has several limitations. 1. The age distribution in the ATAG data set with a preponderance of subjects between 20–30 and 50 years and older but only few middle-aged participants did not allow for more sophisticated modelling of potential age effects. 2. The ATAG data repository does not provide cognitive or behavioral data and so it is not possible to confirm that the layer-specific atrophy impacts episodic memory or working memory and attention. 3. The susceptibility artifacts in regions close to air filled spaces are more pronounced at 7T and resulted in many cases in a sub-optimal signal in the inferior/medial temporal regions. It cannot be excluded that this has influenced the findings in these regions. 4. The MP2RAGE images used in this study came from a publicly available data set that focused on subcortical structures and accordingly used parameters optimized to enhance contrasts in these regions. It cannot be excluded that sequence parameters optimized to enhance cortical rim contrasts would have produced better results, e.g., segmentation of a transition zone corresponding to layer 3 and 4. This needs to be investigated in future work. 5. The heavily/sparsely myelination

maps were compared to histological preparations in the literature. Future work should try to confirm the assumptions made in the discussion section with ex vivo MR images and/or histological preparations. 6. Aging effects on cortical myelin and iron content are complex and layer-specific [14, 53]. Combining the MP2RAGE cortical segmentation approach with additional quantitative modalities, e.g., susceptibility weighted imaging, might provide additional information allowing for a more in-depth interpretation of the findings of this and similar studies.

Taken together, although more work needs to be done, the findings presented here suggest that it is possible to obtain meaningful segmentations of sparsely and heavily myelinated cortical layers using the MP2RAGE sequence. Although originally developed for 7T magnets, the MP2RAGE sequence is also part of the 3T Siemens neuro sequence package. Whole brain submillimeter resolution images can also be acquired on modern 3T magnets which means that the approach presented here is not restricted to high field magnets. Given the evidence that different neurodegenerative diseases have been shown to affect layers differently, e.g., tauopathy in chronic traumatic encephalopathy affects mostly layers 1–3 and in AD mostly layers 5–6 [5], this means that this technique could provide very useful diagnostic information in clinical settings.

## Supporting information

**S1 Table. Atag subjects selected.**
(XLSX)

## Author Contributions

**Conceptualization:** Susanne G. Mueller.

**Formal analysis:** Susanne G. Mueller.

**Methodology:** Susanne G. Mueller.

**Writing – original draft:** Susanne G. Mueller.

**Writing – review & editing:** Susanne G. Mueller.

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
