## [Decision Letter · Decision Letter 0]

13 Sep 2023

PONE-D-23-169427T MP2RAGE for cortical myelin segmentation: Impact of AgingPLOS ONE

Dear Dr. Mueller,

Thank you for submitting your manuscript to PLOS ONE. After careful consideration, we feel that it has merit but does not fully meet PLOS ONE’s publication criteria as it currently stands. Therefore, we invite you to submit a revised version of the manuscript that addresses the points raised during the review process. I also wanted to apologize for the length of time it has taken to render a decision on your manuscript. For some reason, I encountered substantial difficulty in securing a second reviewer. Regarding the specific comments raised by the Reviewers, both had concerns regarding how the clustering approach utilized. Please ensure that this is adequately addressed in the revision. Please submit your revised manuscript by Oct 28 2023 11:59PM. If you will need more time than this to complete your revisions, please reply to this message or contact the journal office at plosone@plos.org. Please include the following items when submitting your revised manuscript:A rebuttal letter that responds to each point raised by the academic editor and reviewer(s). You should upload this letter as a separate file labeled 'Response to Reviewers'.A marked-up copy of your manuscript that highlights changes made to the original version. You should upload this as a separate file labeled 'Revised Manuscript with Track Changes'.An unmarked version of your revised paper without tracked changes. You should upload this as a separate file labeled 'Manuscript'.

We look forward to receiving your revised manuscript.

Kind regards,

Niels Bergsland

Academic Editor

PLOS ONE

3. We note that Figures 1, 2, 3 and 4 in your submission contain copyrighted images. All PLOS content is published under the Creative Commons Attribution License (CC BY 4.0), which means that the manuscript, images, and Supporting Information files will be freely available online, and any third party is permitted to access, download, copy, distribute, and use these materials in any way, even commercially, with proper attribution. For more information, see our copyright guidelines: http://journals.plos.org/plosone/s/licenses-and-copyright.

1. You may seek permission from the original copyright holder of Figures 1, 2, 3 and 4 to publish the content specifically under the CC BY 4.0 license.

Reviewers' comments:

Reviewer's Responses to Questions

**Comments to the Author**

1. Is the manuscript technically sound, and do the data support the conclusions?

Reviewer #1: Yes

Reviewer #2: Yes

2. Has the statistical analysis been performed appropriately and rigorously? 

Reviewer #1: Yes

Reviewer #2: Yes

3. Have the authors made all data underlying the findings in their manuscript fully available?

Reviewer #1: Yes

Reviewer #2: Yes

4. Is the manuscript presented in an intelligible fashion and written in standard English?

Reviewer #1: Yes

Reviewer #2: Yes

5. Review Comments to the Author

Reviewer #1: The goal of this study was to distinguish densely and sparsely myelinated sections along the cortical ribbon using novel MP2rage-derived contrasts acquired at 7T. To achieve this, the author applied k-means clustering to various combination of images and through this pure data-driven approach, they showed that contrast between high and low myelin areas could be enhanced, and the resulting segmentations were visually comparable to histology. To further explore its application potential, the author showed that separating densely vs. sparsely myelinated cortical portions provided unique information characterizing cortical changes in aging.

This study has a number of strengths in terms of imaging technique and statistical analysis, and the manuscript itself was concise and very well written. Nonetheless, I believe the paper could benefit from further refinement and there are a few areas in which the methodology is unclear as described. These concerns are detailed below.

1. The aim of this study is to separate the cortex into high vs. low myelinated “layers”. In the context of k-means clustering, the simplest implementation is to find two clusters in all cortical grey matter voxels. I’m therefore curious why the author chose to start with whole brain maps and set out to find 6 clusters instead. As a result, the current approach required several clean-up steps before arriving at the final cortical segmentations (described in 2.3.2-2.3.3). Please describe the rationale behind the current k-means approach.

2. The author mentioned that segmentations generated from INVC and RATIO image only was noisier compared to multi-contrast combinations. When comparing different multi-contrast combinations, it was further shown that the combination of UNI, T1map, and RATIO images (all but INVC) provided better clustering results. I would therefore conclude that INVC, despite appearing to have high contrast inside cortical ribbon, is not very informative at separating myelinated areas. However, INVC was the only image used in the aging part of the study. Please justify this choice and describe in particular why RATIO is not used over INVC.

3. The B1+ inhomogeneity can be much larger at 7T so the SPM bias correction within the unified segmentation routine might fail. Please include more details about this step and discuss whether voxel signal intensities may be differentially affected by this correction which in turn affecting the k-means clustering.

4. The current study proposed a new way of enhancing contrast between cortical layers that can be used in aging research. More discussion comparing previous studies tackling similar problems is therefore needed for readers to evaluate the robustness and added value of current approach. Please briefly discuss current k-means segmentation results with Rowley, Front Neurosci 2015 (https://pubmed.ncbi.nlm.nih.gov/26557052/), which identified myelinated cortical thickness using T1w image, and current aging results with Sui, Neuroimage 2022 (https://pubmed.ncbi.nlm.nih.gov/36368498/), which looked at depth-specific changes in normal aging.

5. As a limitation, the author should mention that while iron and myelin tend to co-localize in the cortex, which allows T1 to be used as a myelin marker, they can dissociate in older age. Even for healthy aging population, the cortex may have iron accumulation, which has an opposite effect on T1 compared to demyelination.

Minor comments:

- When establishing k-means clustering for cortical segmentation, there seem to be several steps using “randomly selected subjects” (e.g., p.14: “best segmentation… was determined based on the results in 4 randomly selected subjects”). I think a more robust/convincing way might be using all subject or even k-fold cross validation but the author might have a specific reason for doing this. A brief mention of this and potential bias it might introduce may be helpful.

- There are four images in the study so when testing which combinations to use for k-means, there should be 15 combinations to choose from. Is there a particular reason why the current 7 combinations were tested? I think including a combination of UNI and T1map would also be informative but this is up to the author.

- Page 16 (section 2.4): “Finally, the unscaled bias corrected INVC image was used [to] investigate…”

- Page 21: “thinning” misspelled.

Reviewer #2: Strengths:

- Compared to previous works, the quantitative nature could improve transferability

Weaknesses:

- Some of the methods and wording was unclear. I have provided suggestions to help clarify the manuscript.

- Further evidence should be provided to support the methods, or explanations as to why the chosen methods were used. My areas of concern are listed below:

Methods:

- It should be noted that the UNI image is generated by using the complex data from the INV1 and INV2 image (equation 3 in their reference 21 - Marques et al 2010)

- INVC = (INV1 *INV2)/(INV1*INV1) +(INV2*INV2))*100 -> equation provided is missing a ‘(‘ somewhere. I suspect it is supposed to be for the denominator? The author should consider writing this as an inline equation instead for improved readability as their manuscript largely revolves around its calculation.

- Figure 1 -> the equation above for INVC does not match what is in the figure.

- It is not clear why the intensity needs to be adjusted/rescaled. From referencing ref 24, it doesn’t appear to do much asides from sliding the histogram, and possibly bring the intensities onto a common scale (which may be important for Ratio, but not necessarily). Has the author demonstrated that this significantly impacts the results of k-means clustering? This part of the manuscript is convoluted and requires external reading to understand the methods. In my opinion, this complication should only be included if it is essential for the results.

- “The signal intensities of the UNI and T1map image covary with the myelin content but while the UNI signal increases with higher myelinization, the T1 map signal decreases which results in an enhanced contrast between heavily and sparsely myelinated regions in the RATIO image (please see Figure 1).” – the use of language regarding myelination is too strong. The author already acknowledged that iron can also influence T1, so this should be reworded (see also: Stüber C, Morawski M, Schäfer A, Labadie C, Wähnert M, Leuze C, Streicher M, Barapatre N, Reimann K, Geyer S, Spemann D, Turner R. Myelin and iron concentration in the human brain: a quantitative study of MRI contrast. Neuroimage. 2014 Jun;93 Pt 1:95-106. doi: 10.1016/j.neuroimage.2014.02.026. Epub 2014 Mar 6. PMID: 24607447.)

- Can the authors further support the use of 6 clusters by reporting the summed distance to the centroid for varying numbers of clusters? This part is also unclear, what is fed into the k-means clustering? My interpretation of the reading was that the GM was masked with threshold of 0.15, and the resulting masked data is input to k-means. This should primarily leave GM and some partial volume voxels? The description for the use of 6 clusters included clusters for CSF and WM. This doesn’t appear to make sense, and the author should give further explanation and/or proof that 6 clusters is the appropriate number. It appears possible that 4 could be better (remove CSF and WM cluster), but the manuscript stated only 6 to 8 clusters were investigated.

- The subsequent paragraph in the manuscript might suggest that all the rescaling was necessary to set consistent starting guesses for centroid values. Can the author comment on this? How much are the results influenced by the input values? If they are highly influenced, it might also provide support that too many clusters were used.

- “Effects of aging on low and intermediate intensity cortical layers were investigated with surface and volume-based analyses thus allowing for a comparison of the thinning/volume loss patterns and T peaks of the two approaches.” -> low and intermediate intensity is arbitrary as you have stated that UNI and T1 maps have opposing contrasts. The specifics under investigation should be made clearer throughout the manuscript. The author already defined inner cortical and outer cortical layers and should maintain that convention throughout. Also, it is unclear what “T peaks” is from this excerpt. It probably refers to a t-score map from a t-test comparison between young and old, but this should be explicitly stated.

- “Finally, the unscaled bias corrected INVC image was used investigate age-related myelin signal changes” – used to investigate?

- Perhaps a figure as a flowchart is necessary if all steps are deemed essential to the analysis.

- Can the author speak to why they chose to do two age groups (which are quite wide), compared to using a linear model with linear and quadratic age terms? Their other analysis references commonalities with surface-based approaches, so I am unclear why this has also not been maintained.

Results:

- I am not sure if it was a feature of the upload/transfer of files, but the brain images and segmentations in Figure 2 are blurring. Since the segmentations are binary, it would help if they were provided at a higher resolution. In its current state, it is hard to validate the results reported in the figure caption. I did download the figure, but the MRI images and segmentation are blurry.

- Figure 4 – I cannot make sense of the figure. The colour bar goes hot and cold, which should remove the need for the two separate directional comparisons. I.E. everything in the bottom plots should be in the top plots using the cold end of the colormap. Or remove the cold side of the colour map. Or a clearer description needs to be made of what is being presented.

Discussion

- The author keeps referring to cytoarchitecture, but seemingly provided no evidence to support the correlation between that and MR. The MR contrasts seem to primarily follow myeloarchitecture patterns.

- It is known that MP2RAGE is impacted by B1 inhomogeneity. The authors should discuss the potential impact of this on their findings.

Marques, Jose P., and Rolf Gruetter. "New developments and applications of the MP2RAGE sequence-focusing the contrast and high spatial resolution R1 mapping." PloS one 8.7 (2013): e69294.

Haast, Roy AM, et al. "Effects of MP2RAGE B1+ sensitivity on inter-site T1 reproducibility and hippocampal morphometry at 7T." Neuroimage 224 (2021): 117373.

Of particular note, the T map clusters in Figure 3 for the surface analysis appear similar to B1 inhomogeneity maps. Lower intensity layers were significant where B1+ is high, and intermediate layers were significant were B1+ is classically low. Could this be related to the ambiguity in classification of layers 3+ 4 mentioned in discussion?

- The authors used a “b1 correction” algorithm that aims to sharpen the histogram of each tissue type. Does the author believe that this is a necessary step instead of using a B1 map-based correction? It is likely this was done as such due to the data availability, but it is worth a discussion point.

- The work presented in this manuscript appears to be an extension of work previously done using only a T1-weighted contrast. The authors should discuss how their work compares and advances findings over this methodology: Rowley, C. D., Bazin, P. L., Tardif, C. L., Sehmbi, M., Hashim, E., Zaharieva, N., ... & Bock, N. A. (2015). Assessing intracortical myelin in the living human brain using myelinated cortical thickness. Frontiers in neuroscience, 9, 396.

Minor Comments:

MP2RAGE is classically written using all capital letters. I suggest that this convention is maintained in the current manuscript.

6. PLOS authors have the option to publish the peer review history of their article (what does this mean?). If published, this will include your full peer review and any attached files.

Reviewer #1: No

Reviewer #2: No

---

## [Author Response · Author response to Decision Letter 0]

18 Oct 2023

Rebuttal

The author would like to thank the reviewers for their insightful comments. The answers to the comments are written in Arial underneath each comment.

Reviewers' comments:

Reviewer #1: The goal of this study was to distinguish densely and sparsely myelinated sections along the cortical ribbon using novel MP2rage-derived contrasts acquired at 7T. To achieve this, the author applied k-means clustering to various combination of images and through this pure data-driven approach, they showed that contrast between high and low myelin areas could be enhanced, and the resulting segmentations were visually comparable to histology. To further explore its application potential, the author showed that separating densely vs. sparsely myelinated cortical portions provided unique information characterizing cortical changes in aging.

This study has a number of strengths in terms of imaging technique and statistical analysis, and the manuscript itself was concise and very well written. Nonetheless, I believe the paper could benefit from further refinement and there are a few areas in which the methodology is unclear as described. These concerns are detailed below.

1. The aim of this study is to separate the cortex into high vs. low myelinated “layers”. In the context of k-means clustering, the simplest implementation is to find two clusters in all cortical grey matter voxels. I’m therefore curious why the author chose to start with whole brain maps and set out to find 6 clusters instead. As a result, the current approach required several clean-up steps before arriving at the final cortical segmentations (described in 2.3.2-2.3.3). Please describe the rationale behind the current k-means approach.

Answer: Only the re-scaling step used the whole brain. The segmentation step used a SPM generated smoothed gray matter tissue map thresholded at 0.15 (region of interest mask Cortical segmentation: Raw maps) that excluded deep white matter but included superficial white matter. There were two reasons for choosing a region of interest map that also included superficial white matter and sulcal CSF. 1. Tailoring this mask to the cortical gray matter rim by choosing an arbitrary (avoiding gray/white and csf/gray partial volume voxels) threshold would mean that the segmentation of the cortical rim boundaries is largely determined by the performance of the SPM segmentation algorithm and the arbitrary threshold used for its binarization. 2. Although this project only used the cortical segmentations, the other tissue type maps and in particular the gray/white partial volume tissue map are also of interest for disease states associated with a blurring of the gray/white boundary, e.g., subtle cortical dysplasias causing epilepsy. The gray/white partial volume maps were also assessed in this project but as anticipated showed not group differences and therefore the results were not reported. Furthermore, confining the segmentation to the cortical rim would not have eliminated the clean-up steps. One of them was done to eliminate subcortical gray matter structures, e.g. thalamus, hippocampus, i.e., would in this form or another also been necessary if the segmentation would have been limited to the cortical rim. The second clean-up step was necessary to correct misclassified cortical rim voxels in regions that were affected by B1 inhomogeneities due to their closeness to the air-filled spaces, i.e., inferior temporal and orbito-frontal regions. The best way to eliminate or reduce this misclassification is to use dielectric pads in the neck and cheek regions during the image acquisition or to acquire a B1 map for correction in post-processing (Marques and Gruetter, 2013, PMID: 23874936). The atag data does not contain a B1 map and the paper describing it (Ref 23) does not mention the use of dielectric pads. Please see also the answer to reviewer 2’s criticism No 6. The relevant parts of the manuscript were changed as follows (bold) to address the issue raised by the reviewer…

A region of interest map was generated by smoothing the UNI derived gray matter map with a Gaussian smoothing kernel of 2 by 2 by 2 mm FWHM and then binarizing it by thresholding at 0.15. This threshold resulted in a mask that not only contained the gray matter rim but also sulcal/gyral csf and subcortical white matter as well as csf/gray, gray/white transition or partial volume regions. The reasons for choosing this larger mask instead of just a mask focusing on the cortical rim were 1. The cortical rim width is determined by k-means clustering, i.e., does not depend on the performance of the SPM segmentation algorithm. 2. Although not of interest for this project, the gray/white partial volume map may contain important information for diseases characterized by gray/white blurring, e.g., subtle cortical malformations. This mask was used to extract the tissue intensities from each of the subject’s rescaled images, that were converted into image specific z‐scores and then alone (combinations 1 and 2) or combined with the other contrasts (combinations 3-7) supplied to the k‐means clustering algorithm implemented in MATLAB 9.4 (The Math Works, Natick, MA; number of clusters n = 6, squared Euclidian distance function, maximum number of iterations = 1,000, replicates = 100). 

2. The author mentioned that segmentations generated from INVC and RATIO image only was noisier compared to multi-contrast combinations. When comparing different multi-contrast combinations, it was further shown that the combination of UNI, T1map, and RATIO images (all but INVC) provided better clustering results. I would therefore conclude that INVC, despite appearing to have high contrast inside cortical ribbon, is not very informative at separating myelinated areas. However, INVC was the only image used in the aging part of the study. Please justify this choice and describe in particular why RATIO is not used over INVC.

Answer: A more conservative conclusion re usefulness of INVC is that it does not improve the segmentation performance substantially beyond what can already be gained from the combination of UNI, T1map and RATIO image. Furthermore, the INVC was not the only image used in the aging part of the study. This part mostly used the low and intermediate intensity cortical rim maps derived from the combination of UNI, T1map and RATIO. The INVC and the RATIO image both allow to identify low and intermediate intensity zones visually. However in contrast to the heavily processed RATIO (skull-stropped ratio of 2 rescaled, bias corrected image), the INVC image used for the surface analysis has undergone limited processing (bias corrected ratio of two raw images) and that corresponds well to the type of processing that is necessary to create the T1/T2 ratio image. This was the main reason for choosing it over the RATIO image for the surface analysis. Furthermore, the main reason for the different performance of the INVC image in the surface-based analysis compared to the intensity maps was the limited sampling at 0.25/0.5 and 0.75 thickness. This would not have been different if the RATIO image would have been used in the aging analysis. Please see also answer to criticism 4. 

3. The B1+ inhomogeneity can be much larger at 7T so the SPM bias correction within the unified segmentation routine might fail. Please include more details about this step and discuss whether voxel signal intensities may be differentially affected by this correction which in turn affecting the k-means clustering.

Answer: The MP2RAGE sequence was explicitly designed by Marques et al. (Ref 21) to reduce this particular problem. This sequence simultaneously acquires two gradient echo images with different inversion times (INV1 and INV2) but otherwise identical sequence parameters, i.e., B1- , M0, T2* are affected in an identical matter in both images. Combining them by means of a ratio will result in an image (UNI and INVC) that is largely independent from B1- , M0, T2*, i.e., the typical 7T bias field will be greatly reduced, i.e., is considerably less than in a 3T standard MPRAGE which was the reason why the no-ants-n4 flag had to be set for freesurfer processing. Another way to acquire bias field independent images is to perform quantitative imaging which in the case of the MP2RAGE sequence is a T1 relaxation map that can be calculated from the two INV images with the help of a look-up table (please see details in Ref 21). The issue was further investigated by the same authors (Marques and Gruetter, 2013 PLoS One . 2013 Jul 16;8(7):e69294.) who demonstrated that residual B1 inhomogeneities can be removed using the information of a B1 map. The atag data repository does not contain a whole brain B1 map and thus SPM was used to remove these residual B1 inhomogeneities. The following wording was added in “Imaging” first paragraph. to address the reviewer’s concern:

The MP2RAGE sequence generates an INV1 or TI1 gradient echo image, and an INV2 or TI2 gradient echo image. These two images can be used to calculate three additional images 1. A synthetic T1 weighted image (UNI) that is derived from the complex INV1 and INV2 data. 2. A T1 relaxation map (T1map) that can be calculated from the INV1 and INV2 images with a protocol specific look-up table (21). 3. INV1/INV2 ratio image ( INVC) that is generated by combining INV1 and INV2 using the following formula: see Figure 1).

INVC=(INV1*INV2)/((INV1*INV1)+(INV2*INV2) )*100

Because the INV1 and INV2 are acquired under the same conditions B1- , M0, T2* are affected in an identical matter, and therefore the UNI, T1map and INVC images derived from combining them are largely free proton density contrast, T2 contrast, reception bias field and transit field inhomogeneities.

4. The current study proposed a new way of enhancing contrast between cortical layers that can be used in aging research. More discussion comparing previous studies tackling similar problems is therefore needed for readers to evaluate the robustness and added value of current approach. Please briefly discuss current k-means segmentation results with Rowley, Front Neurosci 2015 (https://pubmed.ncbi.nlm.nih.gov/26557052/), which identified myelinated cortical thickness using T1w image, and current aging results with Sui, Neuroimage 2022 (https://pubmed.ncbi.nlm.nih.gov/36368498/), which looked at depth-specific changes in normal aging.

Answer: The study of Sui et al. and their findings that are similar to the findings of this study are now mentioned in the Discussion because it further illustrates the challenges encountered with the T1/T2 ratio-based surface based approach. The study of Rowley et al is mentioned in the introduction but not discussed in detail in the Discussion section for the following reasons. 1. As mentioned in the introduction, several techniques for the imaging of cortical layers have been proposed and some of them, e.g., Viviani et al. also used a “hard” segmentation approach to split the cortex into a sparsely and heavily myelinated layers similarly as Rowley et al. and this study. Discussing them all re robustness is beyond the scope of this manuscript and doing it in a meaningful way would require applying all these techniques to the same population. 2. Rowley et al. used the method to investigate myelination patterns in dipolar disorder and not aging. That prevents comparing its findings with the findings of this papers. Interestingly the same group used an intensity approach, i.e., assessed T1 intensity at 0.25/0.5/0.75 thickness instead of the segmentation approach proposed in the 2015 paper, in a follow-up publication 2017 (PMID: 28462512) despite having acquired the same kind of high quality T1 data.

5. As a limitation, the author should mention that while iron and myelin tend to co-localize in the cortex, which allows T1 to be used as a myelin marker, they can dissociate in older age. Even for healthy aging population, the cortex may have iron accumulation, which has an opposite effect on T1 compared to demyelination.

Answer: The reviewer raises an important issue that has to be kept in mind when interpreting the findings of studies investigating age-associated myelin changes. The current limitation section focus on limitations of this particular manuscript. The age-associated dissociation mentioned by the reviewer is a feature that impacts not only this study but all studies investigating age and also disease related changes of the myelin signal. A multimodal approach, e.g., quantitative susceptibility mapping, would allow for a better insight into the mechanisms driving the changes observed in this study. The following limitation has been added.

6. Aging effects on cortical myelin and iron content are complex and layer-specific (53). Combining the MP2RAGE cortical segmentation approach with additional quantitative modalities, e.g., susceptibility weighted imaging, might provide additional information allowing for a more in-depth interpretation of the findings of this study.

Minor comments:

- When establishing k-means clustering for cortical segmentation, there seem to be several steps using “randomly selected subjects” (e.g., p.14: “best segmentation… was determined based on the results in 4 randomly selected subjects”). I think a more robust/convincing way might be using all subject or even k-fold cross validation but the author might have a specific reason for doing this. A brief mention of this and potential bias it might introduce may be helpful.

Answer: As outlined in the paper, it was only the first step that used a visual assessment of the segmentation quality in 4 subjects. The RATIO and INVC images in which the low and intermediate intensity zones are depicted and show a good correspondence with histological preparations were used as “ground truth” in this step. It was complemented by a quantitative approach (dice) to obtain an objective measure of the differences observed by the visual inspection. The purpose was to identify the image combination that identified the two cortical intensity zones best. A purely data driven approach as proposed by the reviewer would not have answered that question.

- There are four images in the study so when testing which combinations to use for k-means, there should be 15 combinations to choose from. Is there a particular reason why the current 7 combinations were tested? I think including a combination of UNI and T1map would also be informative but this is up to the author.

Answer: The reasons for not investigating the UNI/T1map combination were the same as the reasons for investigating the RATIO and INVC images but not the UNI and T1map as sole input for the cluster analysis. The RATIO and INVC images clearly depicted two cortical intensity zones while the UNI and T1map did not, i.e., it was assumed that the prominent INVC or RATIO contrasts “drive” the segmentation results while the UNI and T1map intensities provide confirmatory information that reduces the noise.

- Page 16 (section 2.4): “Finally, the unscaled bias corrected INVC image was used [to] investigate…”

Answer: corrected

- Page 21: “thinning” misspelled.

Answer: corrected.

Reviewer #2: Strengths:

- Compared to previous works, the quantitative nature could improve transferability

Weaknesses:

- Some of the methods and wording was unclear. I have provided suggestions to help clarify the manuscript.

- Further evidence should be provided to support the methods, or explanations as to why the chosen methods were used. My areas of concern are listed below:

Methods:

1. It should be noted that the UNI image is generated by using the complex data from the INV1 and INV2 image (equation 3 in their reference 21 - Marques et al 2010)

Answer: The information re calculation of the UNI image and the T1 map was added.

2. INVC = (INV1 *INV2)/(INV1*INV1) +(INV2*INV2))*100 -> equation provided is missing a ‘(‘ somewhere. I suspect it is supposed to be for the denominator? The author should consider writing this as an inline equation instead for improved readability as their manuscript largely revolves around its calculation.

Answer: The formula was corrected and all formulas have been written as inline equations.

3. Figure 1 -> the equation above for INVC does not match what is in the figure.

Answer: INV1/INV2 has been replaced by the text “Combination INV1 and INV2”.

4. It is not clear why the intensity needs to be adjusted/rescaled. From referencing ref 24, it doesn’t appear to do much asides from sliding the histogram, and possibly bring the intensities onto a common scale (which may be important for Ratio, but not necessarily). Has the author demonstrated that this significantly impacts the results of k-means clustering? This part of the manuscript is convoluted and requires external reading to understand the methods. In my opinion, this complication should only be included if it is essential for the results.

Answer: The re-scaling section(2.3.1) has been re-written and an additional Figure and Table have been added to explain how the re-scaling works. 

5. “The signal intensities of the UNI and T1map image covary with the myelin content but while the UNI signal increases with higher myelinization, the T1 map signal decreases which results in an enhanced contrast between heavily and sparsely myelinated regions in the RATIO image (please see Figure 1).” – the use of language regarding myelination is too strong. The author already acknowledged that iron can also influence T1, so this should be reworded (see also: Stüber C, Morawski M, Schäfer A, Labadie C, Wähnert M, Leuze C, Streicher M, Barapatre N, Reimann K, Geyer S, Spemann D, Turner R. Myelin and iron concentration in the human brain: a quantitative study of MRI contrast. Neuroimage. 2014 Jun;93 Pt 1:95-106. doi: 10.1016/j.neuroimage.2014.02.026. Epub 2014 Mar 6. PMID: 24607447.)

Answer: Please see also Answer to criticism 5. The section has been extensively re-written and the statement has been deleted.

6. Can the authors further support the use of 6 clusters by reporting the summed distance to the centroid for varying numbers of clusters? This part is also unclear, what is fed into the k-means clustering? My interpretation of the reading was that the GM was masked with threshold of 0.15, and the resulting masked data is input to k-means. This should primarily leave GM and some partial volume voxels? The description for the use of 6 clusters included clusters for CSF and WM. This doesn’t appear to make sense, and the author should give further explanation and/or proof that 6 clusters is the appropriate number. It appears possible that 4 could be better (remove CSF and WM cluster), but the manuscript stated only 6 to 8 clusters were investigated.

Answer: The GM mask was smoothed map with a Gaussian smoothing kernel of 2 by 2 by 2 mm FWHM before thresholding it a 0.15. As a consequence the mask not only contained cortical rim voxels and partially volumed csf/gray and gray/white voxels but also pure subcortical white matter and sulcal csf. There were two reasons for creating this larger mask. 1. Tailoring this mask to the cortical gray matter rim would mean that the segmentation of cortical rim is largely determined by the performance of the SPM segmentation algorithm and the arbitrary threshold chosen for its binarization instead of the k-means clustering. 2. This project only used the cortical segmentations. However, the other tissue type maps and in particular the gray/white partial volume tissue map are also of interest for disease states associated with a blurring of the gray/white boundary, e.g., subtle cortical dysplasia causing epilepsy. The gray/white partial volume maps were also assessed in this project but as anticipated showed not group differences and therefore the results were not reported. The pure white and CSF maps finally allowed for a better segmentation of the partial volume maps. Taken together, the choice of 6 clusters was based on biological considerations rather than cluster centroid distances. The reason for trying out 7 and 8 clusters to find out if this would identify reliably an intermediate/low intensity partial volume zone which was not the case. The reason for trying out 3-5 clusters was to investigate how well the approach identifies the three main tissue types (3 clusters) while cluster 4 and 5 were mostly tested to learn more about how the approach identifies partial volumed clusters. 

7. The subsequent paragraph in the manuscript might suggest that all the rescaling was necessary to set consistent starting guesses for centroid values. Can the author comment on this? How much are the results influenced by the input values? If they are highly influenced, it might also provide support that too many clusters were used.

Answer: Please also see answer to criticism 4. As mentioned in “Cortical Segmentation: Raw Maps”. the intensities of each image type were standardized by converting them to z-scores which also ensured that the intensities from each images were weighted equally.

8. “Effects of aging on low and intermediate intensity cortical layers were investigated with surface and volume-based analyses thus allowing for a comparison of the thinning/volume loss patterns and T peaks of the two approaches.” -> low and intermediate intensity is arbitrary as you have stated that UNI and T1 maps have opposing contrasts. The specifics under investigation should be made clearer throughout the manuscript. The author already defined inner cortical and outer cortical layers and should maintain that convention throughout. Also, it is unclear what “T peaks” is from this excerpt. It probably refers to a t-score map from a t-test comparison between young and old, but this should be explicitly stated.

Answer: The wording intermediate and low intensity zone was based on the appearance of the two zones in the INVC and RATIO images (please see Figure 1) as pointed out in “Introduction” and repeated in the first paragraph of “Conclusion. The text has been edited so that the terms low and intermediate intensity zones are consistently used when referring to the MR images.

The cited sentence was modified as follows for clarification:

Effects of aging on low and intermediate intensity cortical layers were investigated using t-tests (young adults > older adults) with surface- and volume-based analyses to allow for a comparison of the thinning/volume loss patterns and T peaks of the two approaches.

9. “Finally, the unscaled bias corrected INVC image was used investigate age-related myelin signal changes” – used to investigate?

Answer: The sentence has been corrected.

9. Perhaps a figure as a flowchart is necessary if all steps are deemed essential to the analysis.

Answer: It is not entirely clear if the reviewer is asking for a flow chart showing the image processing/segmentation steps or a flow chart summarizing the analysis of the resulting maps in SPM (voxel-based) and Freesurfer (surface-based). A new figure showing the processing steps and the use of the outputs for the analyses but without the processing steps in the analyses was added.

10. Can the author speak to why they chose to do two age groups (which are quite wide), compared to using a linear model with linear and quadratic age terms? Their other analysis references commonalities with surface-based approaches, so I am unclear why this has also not been maintained.

Answer: The reason for this has been mentioned in the limitation section: 

The age distribution in the ATAG data set with a preponderance of subjects between 20 - 30 and 50 years and older but only few middle-aged participants did not allow for more sophisticated modelling of potential age effects. 

In detail, there were 27 patients aged 19-28, mean(SD): 23.8 (2.4) years and 18 older adults aged 42-73, mean(SD) 60.5 9.8 years. There were only two subjects (42 and 45 years) in the 4th and 5th decade

Results:

11. I am not sure if it was a feature of the upload/transfer of files, but the brain images and segmentations in Figure 2 are blurring. Since the segmentations are binary, it would help if they were provided at a higher resolution. In its current state, it is hard to validate the results reported in the figure caption. I did download the figure, but the MRI images and segmentation are blurry.

Answer: The image formatting required by PLOS ONE required a down-sampling of all figures. An improved version of the Figure has been uploaded.

12. Figure 4 – I cannot make sense of the figure. The colour bar goes hot and cold, which should remove the need for the two separate directional comparisons. I.E. everything in the bottom plots should be in the top plots using the cold end of the colormap. Or remove the cold side of the colour map. Or a clearer description needs to be made of what is being presented.

Answer: The color bar has been adapted to show only thinning in red.

Discussion

13. The author keeps referring to cytoarchitecture, but seemingly provided no evidence to support the correlation between that and MR. The MR contrasts seem to primarily follow myeloarchitecture patterns.

Answer: Please see Figure 3 for evidence how myeloarchitecture and cytoarchitecture relate to each other and why the first might allow for assumptions re composition of the latter.

14. It is known that MP2RAGE is impacted by B1 inhomogeneity. The authors should discuss the potential impact of this on their findings. Marques, Jose P., and Rolf Gruetter. "New developments and applications of the MP2RAGE sequence-focusing the contrast and high spatial resolution R1 mapping." PloS one 8.7 (2013): e69294. Haast, Roy AM, et al. "Effects of MP2RAGE B1+ sensitivity on inter-site T1 reproducibility and hippocampal morphometry at 7T." Neuroimage 224 (2021): 117373.

Of particular note, the T map clusters in Figure 3 for the surface analysis appear similar to B1 inhomogeneity maps. Lower intensity layers were significant where B1+ is high, and intermediate layers were significant were B1+ is classically low. Could this be related to the ambiguity in classification of layers 3+ 4 mentioned in discussion?

Answer: As stated by Marques et al. the best way to reduce B1 inhomogeneities is to acquire a B1 map or to use dielectric pads. The atag project did not acquire B1 maps and the paper describing the repository does not mention the use of dielectric pads. Regions where B1 was too low to meet adiabatic conditions affected the mid-temporal inferior temporal gyrus (similarly as shown in Fig 7 of Marques et al.) and were more prominent on the right side. As mentioned in the manuscript (Cortical Segmentation: Final maps), brain regions above the os petrosum and sinus sphenoidalis are also affected by B1 inhomogeneities which impact the segmentation performance and required an additional clean-up step. This has been mentioned in the manuscript in the discussion as limitation (No 3) and acknowledges that this might prevent the detection of findings in these regions. The localization of these effects depends on the coil and as shown by Marques et al. on the head positioning and head size/shape. According to the atag paper, the same coil was used for all participants. Head positioning is usually standardized in research institutions and the images suggest that this was also the case for the images acquired for the atag data repository. It seems unlikely that the low intensity findings in Figure 3 are driven by B1 inhomogeneities because a. in contrast to the inferior temporal gyrus, the posterior parahippocampal gyrus was not affected by these inhomogeneities. b. It can be assumed that the B1 inhomogeneity pattern is similar in both groups because it seems unlikely that one group had a systematically larger head or different head shape that would have resulted in a significant group difference in the B1 inhomogenity fields.

15. The authors used a “b1 correction” algorithm that aims to sharpen the histogram of each tissue type. Does the author believe that this is a necessary step instead of using a B1 map-based correction? It is likely this was done as such due to the data availability, but it is worth a discussion point.

Answer: It is assumed the reviewer refers to the use of the SPM bias correction algorithm that removes slowly varying intensity inhomogeneities. It mostly affected intensities of “deep” brain structures, e.g. deep white matter or deep gray matter structures, e.g., thalamus. Since the gray matter mask included deep gray matter structures using SPM for additional bias correction was deemed beneficial but was not formally tested. 

16. The work presented in this manuscript appears to be an extension of work previously done using only a T1-weighted contrast. The authors should discuss how their work compares and advances findings over this methodology: Rowley, C. D., Bazin, P. L., Tardif, C. L., Sehmbi, M., Hashim, E., Zaharieva, N., ... & Bock, N. A. (2015). Assessing intracortical myelin in the living human brain using myelinated cortical thickness. Frontiers in neuroscience, 9, 396.

Answer: Please see also answer to reviewer 1’s criticism 4. The paper has now been added to the reference list together with several other approaches to segment the cortical rim. Although there are commonalities, e.g., “hard” or binary segmentation of heavily and sparsely myelinated rim zones, there are also many differences, e.g., combination of several 3T BRAVO sequences with 1 mm resolution, fuzzy c-clustering with probabilistic segmentation that required arbitrary thresholding, 3 clusters (white matter, myelinated GM and unmyelinated GM) without partial volume and or CSF clusters which means that the CSF/GM boundary was determined by the MDGM segmentation algorithm and not by the fuzzy c-clustering, application in dipolar disorder etc. While it is possible to list all these differences it is difficult to meaningfully comment on the weaknesses and strengths of two quite different approaches and would raise the question why this paper was chosen and not others, e.g. Viviani et al.. A comprehensive review of all the approaches mentioned in the Introduction is better suited for a review paper on this topic though.

Minor Comments:

MP2RAGE is classically written using all capital letters. I suggest that this convention is maintained in the current manuscript.

This has been corrected

---

## [Decision Letter · Decision Letter 1]

4 Dec 2023

PONE-D-23-16942R17T MP2RAGE for cortical myelin segmentation: Impact of AgingPLOS ONE

Dear Dr. Mueller,

Thank you for submitting your manuscript to PLOS ONE. After careful consideration, we feel that it has merit but does not fully meet PLOS ONE’s publication criteria as it currently stands. Therefore, we invite you to submit a revised version of the manuscript that addresses the points raised during the review process. While both Reviewers found that the manuscript was improved as a result of the initial revision, one of the Reviewers has requested additional modifications, which I agree with. Also, please carefully check the manuscript for typographical/grammatical errors throughout. Please submit your revised manuscript by Jan 18 2024 11:59PM. If you will need more time than this to complete your revisions, please reply to this message or contact the journal office at plosone@plos.org. Please include the following items when submitting your revised manuscript:A rebuttal letter that responds to each point raised by the academic editor and reviewer(s). You should upload this letter as a separate file labeled 'Response to Reviewers'.A marked-up copy of your manuscript that highlights changes made to the original version. You should upload this as a separate file labeled 'Revised Manuscript with Track Changes'.An unmarked version of your revised paper without tracked changes. You should upload this as a separate file labeled 'Manuscript'.

We look forward to receiving your revised manuscript.

Kind regards,

Niels Bergsland

Academic Editor

PLOS ONE

Reviewers' comments:

Reviewer's Responses to Questions

**Comments to the Author**

1. If the authors have adequately addressed your comments raised in a previous round of review and you feel that this manuscript is now acceptable for publication, you may indicate that here to bypass the “Comments to the Author” section, enter your conflict of interest statement in the “Confidential to Editor” section, and submit your "Accept" recommendation.

Reviewer #1: All comments have been addressed

Reviewer #2: (No Response)

2. Is the manuscript technically sound, and do the data support the conclusions?

Reviewer #1: Yes

Reviewer #2: Partly

3. Has the statistical analysis been performed appropriately and rigorously? 

Reviewer #1: Yes

Reviewer #2: Yes

4. Have the authors made all data underlying the findings in their manuscript fully available?

Reviewer #1: Yes

Reviewer #2: Yes

5. Is the manuscript presented in an intelligible fashion and written in standard English?

Reviewer #1: Yes

Reviewer #2: Yes

6. Review Comments to the Author

Reviewer #1: The manuscript has significantly improved in this revision. The reformulated equations and new figures are particularly helpful in clarifying the method. All of my comments have been constructively addressed and I have no other concerns (one typo found, see below). Congratulations to the author.

- Right before “Image Processing”: …therefore the three images derived from combining them are largely free [of] proton density contrast…

Reviewer #2: I appreciate the time the author took to respond to my initial concerns and have addressed a few of them in this revision. The inline equations and flow chart have helped to clarify the methods. However, I found some of the responses to not fully address other points. I have listed them below:

At a high-level, the present manuscript is suggesting to segment the cortical grey matter into superficial and deep layers using a k-means clustering approach. This intended outcome is equivalent to Rowley et al 2015, but the approach to get there is different, in part due to having different data.

The author has stated in this revision as to some reasons that they may feel that their new method is superior to the previous method, and it may very well be. It appears the dilation of the GM mask and subsequent re-segmentation with extra clusters could add robustness to the current method. Add to this that the method is intending to use a quantitative metric that should remove additional variance in the image, hopefully increasing its transferability. All these things however appear to be small methodological improvements on a previously reported method with the same application. This comes to light as the last sentence in the first paragraph of the discussion presents this as a completely new method: “The next paragraphs will discuss the new method and the findings in more detail.”

The general comment is that the author includes a lot of introduction and discussion on points that the MRI resolution is unable to resolve, such as the in-depth discussion on the components of different cortical layers. This space would be better utilized discussing how this method relates and improves upon previous ones. Ultimately, this is a manuscript that is proposing a new method, with the aging-information just providing evidence of its utility. At minimum, an additional paragraph in the discussion is warranted to compare this current method to others that aim to do this exact thing. This should also include Viviani et al 2017, as the author mentions, which instead uses Gaussian mixture modeling, but adds in additional contrasts to improve the segmentation. The author needs to provide a critical comparison of their method to the others, otherwise, why did they not use what was previously developed? This question needs to be addressed in the manuscript.

Abstract:

“Myelin and iron are major contributor to” – plural

Introduction:

“Cortical myelin and iron, that often co-localizes with cortical myelin, represent the main components of the cortical MR signal (11-13)” – rewrite

“The second aim was to compare the resulting segmentations with histological myelin maps from the literature (4) to confirm that the two zones indeed represent heavily and sparsely myelinated” – given the data is from separate individuals, you are unable to ‘confirm’ this. You are providing support that it is likely that. This is mentioned in the limitations in the discussion/conclusion but should also me clarified here.

Methods:

Figure 2 caption – “e. Raw segmentations produced by c-means clustering. f. final segmentations after clean-up” - k-means?

In the pre-processing section, the author should cite the paper for the unified segmentation method - J. Ashburner and K.J. Friston. Unified segmentation. NeuroImage, 26:839–851, 2005.

I still believe that the scaling is adding unnecessary steps to this method. The crucial point that needs to be addressed appears to be: “is the absolute value of the images important for the processing”.

The current methodology is simply applying a linear scale factor, globally across the image. Ref1 and Ref2 are each a single number that was empirically determined. The CSF and WM mode values are also single values. So, the resulting scaling factor is a single value that is applied across the whole image. The histograms in Figure 3A and Figure3B are identical if you ignore the numbers on the x-axis. The images that are displayed in Figure 3A and B would also look equivalent if the window and level values set for the 3A portion were scaled by the values used to rescale the image. So overall, this adds an additional step that changes nothing but the values globally in the image. The peaks pointed out do move, but this should give you a RATIO histogram with the same distribution that is scaled by (UNIfact/T1fact). This is why the contrast is unchanged in the image, but the range is ‘flattened’, as the author states. In the first review, I asked for the author to provide a reason for the rescaling of images, and it is possibly clearer now that it is unnecessary. Which means that this statement would not true:

“This had a major impact on the contrast behavior of the RATIO image calculated from these two images. Compared to the RATIO image calculated from the non-rescaled images whose contrast behavior was similar to the UNI image, the image calculated from the re-scaled had an enhanced and “inverted” contrast between heavily and sparsely myelinated regions”

If I am incorrect, then the author should add the histogram for the RATIO image to Figure 3 so that the reader can see this, so they don’t have the same confusion as I have. Referencing their previous work (ref 25), they also displayed histograms for rescaled UNI and T1 maps. Interestingly, in [25] there was also a shift value which appears to have been dropped in this work. The shift is also a global shift, so even it was applied, the scaled ratio would have the same shape with global change in values of ([UNIfact+UNIshift] / [T1fact+T1shift]). The increase in dynamic range could be useful for presetting values for visualization software, or for limiting rounding problems, if the data is strictly forced to be integer values. However, given the histogram shape doesn’t change, and the relative difference between tissue types also goes unchanged, it should not impact the results of the k-means clustering if the data lives in Matlab in double format (standard for matlab processing). I think this is important to work out, as the author is claiming that this is one of their novel contributions to the advancement of being able to categorize the cortex into two layers with MRI.

It is also worth noting that the UNI image by definition should range from -0.5 to 0.5 (see Marques et al 2010), which was partially done to have a standardized way to set color bar limits for visualization (see my point above). It appears the author is using a version from the scanner that is scaled from 0-4096. But this is an important point to bring up here, as the argument I am making is that the absolute values of the metrics moving forwards from this step should not matter, and thus the rescaling is unnecessary. If the absolute value of the resulting image is necessary for the subsequent steps, then this range should be fixed (i.e. from -0.5 to 0.5). Otherwise, the current scaling method will not move negative values onto the positive axis, unless the shift term in [25] is included here. It also appears that the next step of the processing is to z-score these values, which further suggests that this global scaling measure in not needed.

It is also suggested to use larger font in the histograms so that the readers know what data range that their values should be in, if they are trying to replicate this work.

Discussion:

“the potential to provide layer-specific cortical volumetric information”- Needs to be reworded or be made clearer, as you cannot provide layer-specific information for a 6(+) layer tissue, with only two clustered layers. Additionally, the author provides discussion on this exact point of resolution and the ambiguity of where layers 3 and 4 end up yet starts the following paragraph with: “The new layer specific approach’s potential…”.

Minor:

Many locations the author listed “k-mean” but I believe it should be “k-means”.

7. PLOS authors have the option to publish the peer review history of their article (what does this mean?). If published, this will include your full peer review and any attached files.

Reviewer #1: No

Reviewer #2: No

---

## [Author Response · Author response to Decision Letter 1]

20 Dec 2023

Rebuttal

Again, the author would like to thank the reviewers for their insightful comments.

Reviewer #1: 

Right before “Image Processing”: …therefore the three images derived from combining them are largely free [of] proton density contrast…

Answer: The typo has been corrected.

Reviewer #2: 

I appreciate the time the author took to respond to my initial concerns and have addressed a few of them in this revision. The inline equations and flow chart have helped to clarify the methods. However, I found some of the responses to not fully address other points. I have listed them below:

1. At a high-level, the present manuscript is suggesting to segment the cortical grey matter into superficial and deep layers using a k-means clustering approach. This intended outcome is equivalent to Rowley et al 2015, but the approach to get there is different, in part due to having different data.

The author has stated in this revision as to some reasons that they may feel that their new method is superior to the previous method, and it may very well be. It appears the dilation of the GM mask and subsequent re-segmentation with extra clusters could add robustness to the current method. Add to this that the method is intending to use a quantitative metric that should remove additional variance in the image, hopefully increasing its transferability. All these things however appear to be small methodological improvements on a previously reported method with the same application. This comes to light as the last sentence in the first paragraph of the discussion presents this as a completely new method: “The next paragraphs will discuss the new method and the findings in more detail.”

Answer: As pointed out by the reviewer, there was no formal comparison between the method proposed here and those proposed by Rowley et al. 2015 and Viviani et al 2017. So it is assumed that the reviewer refers to the comments in the rebuttal letter. This section only listed the differences between the method introduced by Rowley and in no way mentioned that one method is better than the other. In contrary, it explicitly stated that a meaningful comparison, i.e., a comparison that would include an evidence based judgement of superior performance, is not possible without either at least being able to work with the same study population and/or having the ground truth information, i.e., histological confirmation, for each data set. This is not possible with the available data. Finally, the last remark re “new method” depends on a person’s definition of new because there exist no objective benchmarks for what has to be deemed a new or just an alternative method. Now that short summaries of the methods proposed by Rowley et al. and Viviani et. al. have been added, the readers can decide for themselves if the method proposed here is sufficiently different to be considered new or just an alternative. 

2. The general comment is that the author includes a lot of introduction and discussion on points that the MRI resolution is unable to resolve, such as the in-depth discussion on the components of different cortical layers. This space would be better utilized discussing how this method relates and improves upon previous ones. Ultimately, this is a manuscript that is proposing a new method, with the aging-information just providing evidence of its utility. At minimum, an additional paragraph in the discussion is warranted to compare this current method to others that aim to do this exact thing. This should also include Viviani et al 2017, as the author mentions, which instead uses Gaussian mixture modeling, but adds in additional contrasts to improve the segmentation. The author needs to provide a critical comparison of their method to the others, otherwise, why did they not use what was previously developed? This question needs to be addressed in the manuscript.

Answer: A short discussion of the methods proposed by Rowley et al and Viviani et al has been added to the discussion. A stated before, it is not possible to provide a meaningful comparison of the three methods beyond listing the technical differences. As to the last question, why not use a previously developed method? This is an interesting question particularly in a field that has at least 6 different T1 based methods to measure cortical thickness (FreeSurfer, CIVET, Brainvoyager, Brainsuite, CAT, ANTS) that are all used by the community.

3. Abstract:

“Myelin and iron are major contributor to” – plural

Answer: The typo has been corrected.

4. Introduction:

“Cortical myelin and iron, that often co-localizes with cortical myelin, represent the main components of the cortical MR signal (11-13)” – rewrite

Answer: The statement has been re-written

5. “The second aim was to compare the resulting segmentations with histological myelin maps from the literature (4) to confirm that the two zones indeed represent heavily and sparsely myelinated” – given the data is from separate individuals, you are unable to ‘confirm’ this. You are providing support that it is likely that. This is mentioned in the limitations in the discussion/conclusion but should also me clarified here.

Answer: The sentence has been re-written.

6. Methods:

Figure 2 caption – “e. Raw segmentations produced by c-means clustering. f. final segmentations after clean-up” - k-means?

Answer: The typo has been corrected

7. In the pre-processing section, the author should cite the paper for the unified segmentation method - J. Ashburner and K.J. Friston. Unified segmentation. NeuroImage, 26:839–851, 2005.

Answer: The reference has been added. It should be pointed out though the “segment” as it is currently implemented in SPM12 has been modified quite a bit since the 2005 publication. 

8. I still believe that the scaling is adding unnecessary steps to this method. The crucial point that needs to be addressed appears to be: “is the absolute value of the images important for the processing”.

The current methodology is simply applying a linear scale factor, globally across the image. Ref1 and Ref2 are each a single number that was empirically determined. The CSF and WM mode values are also single values. So, the resulting scaling factor is a single value that is applied across the whole image. The histograms in Figure 3A and Figure3B are identical if you ignore the numbers on the x-axis. The images that are displayed in Figure 3A and B would also look equivalent if the window and level values set for the 3A portion were scaled by the values used to rescale the image. So overall, this adds an additional step that changes nothing but the values globally in the image. The peaks pointed out do move, but this should give you a RATIO histogram with the same distribution that is scaled by (UNIfact/T1fact). This is why the contrast is unchanged in the image, but the range is ‘flattened’, as the author states. In the first review, I asked for the author to provide a reason for the rescaling of images, and it is possibly clearer now that it is unnecessary. Which means that this statement would not true:

“This had a major impact on the contrast behavior of the RATIO image calculated from these two images. Compared to the RATIO image calculated from the non-rescaled images whose contrast behavior was similar to the UNI image, the image calculated from the re-scaled had an enhanced and “inverted” contrast between heavily and sparsely myelinated regions”

If I am incorrect, then the author should add the histogram for the RATIO image to Figure 3 so that the reader can see this, so they don’t have the same confusion as I have. Referencing their previous work (ref 25), they also displayed histograms for rescaled UNI and T1 maps. Interestingly, in [25] there was also a shift value which appears to have been dropped in this work. The shift is also a global shift, so even it was applied, the scaled ratio would have the same shape with global change in values of ([UNIfact+UNIshift] / [T1fact+T1shift]). The increase in dynamic range could be useful for presetting values for visualization software, or for limiting rounding problems, if the data is strictly forced to be integer values. However, given the histogram shape doesn’t change, and the relative difference between tissue types also goes unchanged, it should not impact the results of the k-means clustering if the data lives in Matlab in double format (standard for matlab processing). I think this is important to work out, as the author is claiming that this is one of their novel contributions to the advancement of being able to categorize the cortex into two layers with MRI.

Answer: The reviewer correctly states that the scaling factor is a single value for each image. In the case of the images displayed in Fig 3, the UNI scale factor is 0.0248 and the T1 map scale factor is 0.0288. Multiplying the image intensities with these scale factors reduces the dynamic range of both images while maintaining the overall shape of the histograms but it should not introduce a shift and it also does not explain why the gray/white ratios in Table 1 are slightly different. This is further supported by the fact that neither the RATIO histograms nor the images (please see enlarged cortical rim sections) generated from the non-scaled and scaled images look the same. Taken together, these differences suggest that the re-scaling does not affect all intensities in the way the reviewer suggested. To illustrate this point the raw segmentations generated from the images depicted in Figure 3 were used to extract the intensities of the 5 tissue maps, i.e., csf/gm partial volume (csf/gm), intermediate intensity gray matter zone (int), low intensity gray matter zone (low), gm/wm partial volume (gm/wm) and subcortical white matter (wm). The histograms of these tissue map intensities were combined with the histogram of the whole image histogram. If the re-scaling affects all tissue intensities in the same way one would expect that their histograms look similar in the non-scaled and scaled image with the range being the only difference. However, this is not the case. The most striking differences are seen for the int histograms. The int histogram of the scaled T1 map image has a higher peak with a steeper decline towards higher intensities than the histogram of the non-scaled T1 map indicating a higher number of int voxels in the lower intensity range in the former. The int histogram of the scaled UNI image also has a higher peak compared to the non-scaled UNI but here the scaling increased the number of voxels in the higher intensity range. As a result, the int voxels of the scaled RATIO image are visually brighter than those in the non-scaled image. The scaling effect on the low histogram is very subtle and thus its intensity behavior in the scaled and non-scaled images similar. The different scaling behavior of the int and low intensity voxels is best illustrated by the fact that they switch positions in the scaled RATIO histogram, i.e., the int histogram is on the left side of the low intensity histogram in the non-scaled RATIO image but on the right side in the scaled RATIO image. Based on this the conclusion that the scaling is necessary and that this is most obvious in the RATIO image seems justified. The main effect of the re-scaling is to improve the segmentation of the low intensity cortical zone.

The reviewer is correct that [25] also calculates an image specific left shift and that this is not done in this project. Table 1 in [25] shows that the combined effect is an increased gray/white contrast which improves the segmentation of internal brainstem structures that have lower gray/white contrasts than cortical gray/subcortical white matter that is already strong in the UNI and T1 maps.

9. It is also worth noting that the UNI image by definition should range from -0.5 to 0.5 (see Marques et al 2010), which was partially done to have a standardized way to set color bar limits for visualization (see my point above). It appears the author is using a version from the scanner that is scaled from 0-4096. But this is an important point to bring up here, as the argument I am making is that the absolute values of the metrics moving forwards from this step should not matter, and thus the rescaling is unnecessary. If the absolute value of the resulting image is necessary for the subsequent steps, then this range should be fixed (i.e. from -0.5 to 0.5). 

Answer: The reviewer’s observation is correct. The UNI image calculated by the wip and product version of the MP2RAGE sequence have a larger dynamic range than the images used by Marques et al. 2010. Please see answer to criticism 8 why the re-scaling is not superfluous.

10. It also appears that the next step of the processing is to z-score these values, which further suggests that this global scaling measure in not needed.

Answer: The three input images (re-scaled UNI, T1 map and RATIO) have different dynamic ranges, i.e., the range of the RATIO image is smaller than that of the T1 map and UNI image which results in unequal variances. K-means clustering puts more weight on variables with smaller variances and z-scoring is a way to prevent that, i.e., z-scoring is used to standardize the data.

11. Discussion:

“the potential to provide layer-specific cortical volumetric information”- Needs to be reworded or be made clearer, as you cannot provide layer-specific information for a 6(+) layer tissue, with only two clustered layers. Additionally, the author provides discussion on this exact point of resolution and the ambiguity of where layers 3 and 4 end up yet starts the following paragraph with: “The new layer specific approach’s potential…”.

Answer: The sentence refers to the “potential”, it does not state that it provides 6 layer information. Furthermore, as the reviewer states themselves, large parts of the discussion discuss the limitations of the intermediate/low intensity zones so if it is assumed that the reader does not just skim the Discussion there should be no misunderstanding that it is not possible to depict the 6 layer structure with this approach. 

Minor:

Many locations the author listed “k-mean” but I believe it should be “k-means”.

Answer: Two instances of k-mean have been corrected to k-means.

---

## [Decision Letter · Decision Letter 2]

3 Jan 2024

PONE-D-23-16942R27T MP2RAGE for cortical myelin segmentation: Impact of AgingPLOS ONE

Dear Dr. Mueller,

Thank you for submitting your manuscript to PLOS ONE. After careful consideration, we feel that it has merit but does not fully meet PLOS ONE’s publication criteria as it currently stands. Therefore, we invite you to submit a revised version of the manuscript that addresses the points raised during the review process.

We look forward to receiving your revised manuscript.

Kind regards,

Niels Bergsland

Academic Editor

PLOS ONE

Reviewers' comments:

Reviewer's Responses to Questions

**Comments to the Author**

1. If the authors have adequately addressed your comments raised in a previous round of review and you feel that this manuscript is now acceptable for publication, you may indicate that here to bypass the “Comments to the Author” section, enter your conflict of interest statement in the “Confidential to Editor” section, and submit your "Accept" recommendation.

Reviewer #2: (No Response)

2. Is the manuscript technically sound, and do the data support the conclusions?

Reviewer #2: No

3. Has the statistical analysis been performed appropriately and rigorously? 

Reviewer #2: Yes

4. Have the authors made all data underlying the findings in their manuscript fully available?

Reviewer #2: Yes

5. Is the manuscript presented in an intelligible fashion and written in standard English?

Reviewer #2: Yes

6. Review Comments to the Author

Reviewer #2: I appreciate the time the author took to respond to my initial concerns and have addressed many of them in this revision. The redraft of Figure 3 I think is great, but it appears to highlight an error in the data processing. I cannot recommend this manuscript for publication until this is addressed.

For the problem, it is best to put numbers to this to explain.

I took approximate mean values for each tissue class (in order of WM, GM/WM, GM, low, int, csg/gm) from this histogram and calculated the results myself. The top row all looks correct (unscaled), and the approximate values I pulled are:

T1: 1250, 1500, 1800, 2200, 3000

UNI: 3200, 2700, 2000, 1200, 700

RATIO: 250, 175, 100, 50, 25

If I calculate the ratio based on UNI/T1*100 values I pulled from the histogram, I get: 256, 180, 111, 54, 23 (within rounding error of the histogram RATIO values). Everything looks in order.

If we move to the second row with the scaled values, the author reported the scaling factor was 0.0248 for UNI, and 0.0288 for T1. If I apply that to the above numbers, I get good agreement in the T1scaled and UNIscaled histograms which have approximate values of (again in order of WM, GM/WM, GM, low, int, csf/gm):

T1: 30, 45, 55, 70, 90

UNI: 80, 65, 45, 30, 10

If you take the ratio of this “UNIscale/T1scale *100”, you get:

My calculated RATIO: 260, 144, 81, 42, 11 -> these are all close to the first set, but IMPORTANTLY are monotonically decreasing. So, it should be impossible to get the characteristic dark ring at the border of the WM/GM.

The values reported for the histogram in Figure 3, which match the image displayed are

Figure 3 bottom RATIO: 40, 20, 5, 15, 20

Two obvious problems, 1. They are no longer monotonically decreasing. And 2. None of them match the very simple mathematical operation that is being reported. Perhaps with this, the author can realize my stance that the scaling operation is not needed. But it appears that another image entirely is being used to generate this contrast.

The only thing I can think of, is that this ratio is actually INV1/INV2, or something along those lines. Or you have subtracted an offset in the calculation of the ratio, and then taken the absolute value, to fold the negative values back around. Either way, the author needs to determine what has gone wrong in their processing scripts, and properly reflect their processing methods in the Methods section. This may impact other areas of the manuscript as well.

Another issue that sticks out to me is the change in tissue distributions in the histogram. The change in relative peak heights seems problematic unless variable bar widths are being used. It looks to me as though more voxels are now being counted as being in the ‘int’ tissue class and less in the “low” on the scaled UNI and T1, than in the original. The same segmentation should be used for both, such that the total voxels belonging to a tissue class are consistent. If they are not, then the claim should be more that the different intensity changes the segmentation, rather than the suggesting the intensity itself is changing.

Minor:

Introduction:

“Cortical myelin and iron, that often co-localizes with cortical myelin, ….” – I mentioned this last review, and it still doesn’t make sense to say that cortical myelin co-localizes with itself. Perhaps you mean to say “Iron, which often co-localizes with cortical myelin,…”.

7. PLOS authors have the option to publish the peer review history of their article (what does this mean?). If published, this will include your full peer review and any attached files.

Reviewer #2: No

---

## [Author Response · Author response to Decision Letter 2]

7 Feb 2024

Rebuttal

The author would like to thank the editor for the chance to submit a third revision and the reviewer for their insightful comments that helped to identify and amend a major issue in the previous versions.

Reviewer #2: I appreciate the time the author took to respond to my initial concerns and have addressed many of them in this revision. The redraft of Figure 3 I think is great, but it appears to highlight an error in the data processing. I cannot recommend this manuscript for publication until this is addressed.

For the problem, it is best to put numbers to this to explain.

I took approximate mean values for each tissue class (in order of WM, GM/WM, GM, low, int, csg/gm) from this histogram and calculated the results myself. The top row all looks correct (unscaled), and the approximate values I pulled are:

T1: 1250, 1500, 1800, 2200, 3000

UNI: 3200, 2700, 2000, 1200, 700

RATIO: 250, 175, 100, 50, 25

If I calculate the ratio based on UNI/T1*100 values I pulled from the histogram, I get: 256, 180, 111, 54, 23 (within rounding error of the histogram RATIO values). Everything looks in order.

If we move to the second row with the scaled values, the author reported the scaling factor was 0.0248 for UNI, and 0.0288 for T1. If I apply that to the above numbers, I get good agreement in the T1scaled and UNIscaled histograms which have approximate values of (again in order of WM, GM/WM, GM, low, int, csf/gm):

T1: 30, 45, 55, 70, 90

UNI: 80, 65, 45, 30, 10

If you take the ratio of this “UNIscale/T1scale *100”, you get:

My calculated RATIO: 260, 144, 81, 42, 11 -> these are all close to the first set, but IMPORTANTLY are monotonically decreasing. So, it should be impossible to get the characteristic dark ring at the border of the WM/GM.

The values reported for the histogram in Figure 3, which match the image displayed are

Figure 3 bottom RATIO: 40, 20, 5, 15, 20

Two obvious problems, 1. They are no longer monotonically decreasing. And 2. None of them match the very simple mathematical operation that is being reported. Perhaps with this, the author can realize my stance that the scaling operation is not needed. But it appears that another image entirely is being used to generate this contrast.

The only thing I can think of, is that this ratio is actually INV1/INV2, or something along those lines. Or you have subtracted an offset in the calculation of the ratio, and then taken the absolute value, to fold the negative values back around. Either way, the author needs to determine what has gone wrong in their processing scripts, and properly reflect their processing methods in the Methods section. This may impact other areas of the manuscript as well.

Another issue that sticks out to me is the change in tissue distributions in the histogram. The change in relative peak heights seems problematic unless variable bar widths are being used. It looks to me as though more voxels are now being counted as being in the ‘int’ tissue class and less in the “low” on the scaled UNI and T1, than in the original. The same segmentation should be used for both, such that the total voxels belonging to a tissue class are consistent. If they are not, then the claim should be more that the different intensity changes the segmentation, rather than the suggesting the intensity itself is changing.

Answer: The reviewer is correct. The unexpected behavior (no longer monotonically decreasing histograms) has been caused by a one letter typo in the Matlab code that meant that the script calculated the ratio between INVC and T1map instead of the ratio between UNI and T1map. The latter indeed behaves as predicted by the reviewer and confirms the reasoning that the scaling is not necessary. The scaling step was therefore eliminated from the processing routine. The next step was then to investigate the performance of the true UNI/T1map image and compare it to that of the accidentally calculated INVC/T1map image. This was done by first visually investigating the impact of the two ratios non-scaled ratios, i.e., INVC/T1map (RATIO) and UNI/T1map (sRATIO) on segmentation performance when used as sole input (RATIO, sRATIO, INVC) and of all combinations of these two contrasts with the remaining contrasts. Combinations that were able to identify the 6 tissue categories (csf, csf-gm partial volume, intermediate intensity gray matter rim zone, low intensity gray matter rim zone, gray matter-white matter partial volume and superficial white matter in all test subjects were further evaluated by calculating the mean within cluster (voxel to centroid of cluster it was assigned) and mean between cluster (voxel to all other cluster centroids) distances. Using this metric the contrast combination RATIO, T1map, UNI performed best, closely followed by RATIO, T1map, sRATIO. The data was reprocessed without re-scaling and all analyses repeated. As expected, the findings did not change. 

The contrast combination of sRATIO, T1map and UNI was able to segment the 6 tissue categories but the map with the low intensity gray matter rim zone contained a prominent outer ring corresponding to dura residuals that was so consistent across subjects that it affected the template creation in DARTEL. Considering the inferior performance of the originally intended contrast sRATIO compared to the mistakenly calculated RATIO, the typo turned out to be serendipitous. The manuscript text was re-written to incorporate the new findings. 

Minor:

Introduction:

“Cortical myelin and iron, that often co-localizes with cortical myelin, ….” – I mentioned this last review, and it still doesn’t make sense to say that cortical myelin co-localizes with itself. Perhaps you mean to say “Iron, which often co-localizes with cortical myelin,…”.

Answer: “that” in this sentence refers only to iron and not also to the myelin in the main sentence. Similarly as ….the man and the dog, that barked at all other dogs, went home… does not mean that the man and the dog are both barking at other dogs. This difference in interpretation was the reason that the author did not understand the issue raised by the reviewer. The sentence was re-written to eliminate any possibility of a misunderstanding.

---

## [Decision Letter · Decision Letter 3]

14 Feb 2024

7T MP2RAGE for cortical myelin segmentation: Impact of Aging

PONE-D-23-16942R3

Dear Dr. Mueller,

We’re pleased to inform you that your manuscript has been judged scientifically suitable for publication and will be formally accepted for publication once it meets all outstanding technical requirements.

Kind regards,

Niels Bergsland

Academic Editor

PLOS ONE

Additional Editor Comments (optional):

Reviewers' comments:

Reviewer's Responses to Questions

**Comments to the Author**

1. If the authors have adequately addressed your comments raised in a previous round of review and you feel that this manuscript is now acceptable for publication, you may indicate that here to bypass the “Comments to the Author” section, enter your conflict of interest statement in the “Confidential to Editor” section, and submit your "Accept" recommendation.

Reviewer #2: All comments have been addressed

2. Is the manuscript technically sound, and do the data support the conclusions?

Reviewer #2: Yes

3. Has the statistical analysis been performed appropriately and rigorously? 

Reviewer #2: Yes

4. Have the authors made all data underlying the findings in their manuscript fully available?

Reviewer #2: Yes

5. Is the manuscript presented in an intelligible fashion and written in standard English?

Reviewer #2: Yes

6. Review Comments to the Author

Reviewer #2: The author has clarified all concerns I had with the methodology, I am happy to know it was an easy fix in the end. Table 1 was a nice addition to this revision to compare the effectiveness of the different contrasts for segmentation.

7. PLOS authors have the option to publish the peer review history of their article (what does this mean?). If published, this will include your full peer review and any attached files.

Reviewer #2: No

---

## [Editor Report · Acceptance letter]

23 Mar 2024

PONE-D-23-16942R3 

PLOS ONE

Dear Dr. Mueller, 

I'm pleased to inform you that your manuscript has been deemed suitable for publication in PLOS ONE. Congratulations! Your manuscript is now being handed over to our production team.

Kind regards, 

on behalf of

Dr. Niels Bergsland 

Academic Editor

PLOS ONE